# Nuclear to cytoplasmic transport is a druggable dependency in MYC-driven hepatocellular carcinoma

Anja Deutzmann [1], Delaney K. Sullivan [1], Renumathy Dhanasekaran [1,11], Wei Li [2,3], Xinyu Chen[1], Ling Tong[1], Wadie D. Mahauad-Fernandez[1], John Bell[4], Adriane Mosley[1], Angela N. Koehler [5,6,7,8], Yulin Li [1,12,13] ✉ & Dean W. Felsher [1,9,10,13] ✉

The MYC oncogene is often dysregulated in human cancer, including hepatocellular carcinoma (HCC). MYC is considered undruggable to date. Here, we comprehensively identify genes essential for survival of MYC$^{high}$ but not MYC$^{low}$ cells by a CRISPR/Cas9 genome-wide screen in a MYC-conditional HCC model. Our screen uncovers novel MYC synthetic lethal (MYC-SL) interactions and identifies most MYC-SL genes described previously. In particular, the screen reveals nucleocytoplasmic transport to be a MYC-SL interaction. We show that the majority of MYC-SL nucleocytoplasmic transport genes are upregulated in MYC$^{high}$ murine HCC and are associated with poor survival in HCC patients. Inhibiting Exportin-1 (XPO1) in vivo induces marked tumor regression in an autochthonous MYC-transgenic HCC model and inhibits tumor growth in HCC patient-derived xenografts. XPO1 expression is associated with poor prognosis only in HCC patients with high MYC activity. We infer that MYC may generally regulate and require altered expression of nucleocytoplasmic transport genes for tumorigenesis.

MYC is a master transcription factor that can regulate the expression of thousands of genes in the genome[1,2]. Overexpression of MYC is thought to contribute to the pathogenesis of over 50% of human cancers[3]. Experimentally, the inhibition of MYC expression reverses tumorigenesis in transgenic mouse cancer models using tetracycline-regulated (Tet) MYC expression systems[4–6]. Therapeutically targeting MYC would have broad clinical impact across many types of human cancer.

Hepatocellular carcinoma (HCC) was the fourth leading cause of cancer deaths worldwide in 2018[7]. The number of new HCC cases each year is projected to increase by 35% by the year 2030[8]. Existing conventional and targeted therapies to date show only limited clinical response in patients with surgically unresectable disease. The proximal MYC network is altered in more than 70% of human HCC[9]. Drugs that target the MYC pathway could be effective to treat HCC. However, identifying small molecules that directly target MYC function has

[1]Division of Oncology, Department of Medicine, Stanford University, Stanford, CA 94305, USA. [2]Center for Genetic Medicine Research, Children's National Hospital, Washington, DC 20012, USA. [3]Department of Genomics and Precision Medicine, George Washington University, Washington, DC 20012, USA. [4]Stanford Genome Technology Center, Stanford University, Stanford, CA 94305, USA. [5]Koch Institute for Integrative Cancer Research at MIT, Massachusetts Institute of Technology, Cambridge, MA 02139, USA. [6]Department of Biological Engineering, Massachusetts Institute of Technology, Cambridge, MA 02139, USA. [7]Center for Precision Cancer Medicine, Massachusetts Institute of Technology, Cambridge, MA 02139, USA. [8]Broad Institute of MIT and Harvard, Cambridge, MA 02142, USA. [9]Department of Pathology, Stanford University, Stanford, CA 94305, USA. [10]Stanford Cancer Institute, Stanford University, Stanford, CA 94305, USA. [11]Present address: Division of Gastroenterology, Department of Medicine, Stanford University, Stanford, CA 94305, USA. [12]Present address: Institute for Academic Medicine, Houston Methodist and Weill Cornell Medical College, Houston, TX 77030, USA. [13]These authors contributed equally: Yulin Li, Dean W. Felsher. ✉e-mail: yli@houstonmethodist.org; dfelsher@stanford.edu

proved challenging[10,11]. As an alternative, the identification of targetable gene products that are MYC synthetic lethal could lead to new and better treatments for HCC and other MYC-associated cancers[12].

Here, we combine a CRISPR-based genome-wide library screen with Tet system-regulated MYC expression to identify gene targets that, when genetically inactivated, are lethal to cancer cells only with high but not low MYC expression. In this work, we comprehensively describe MYC synthetic lethal gene interactions and specifically identify nucleocytoplasmic transport as a therapeutic vulnerability in MYC-driven HCC.

## Results

### CRISPR screen identifies MYC synthetic lethal genes

To identify genes that negatively or positively affect proliferation or survival of MYC$^{high}$ tumor cells but not MYC$^{low}$ normal cells, we have performed a CRISPR genome-wide library screen in a tumor cell line (designated EC4) derived from a Tet system-regulated transgenic mouse model of MYC-induced HCC (*LAP-tTA/tet-O-MYC*/FVB/N)[13]. In this model, human MYC transgene expression can be conditionally shut off by treatment with doxycycline (Supplementary Fig. 1). We constitutively expressed Cas9 nuclease in EC4 cells and confirmed Cas9 to be functional in inducing mutations at specific genomic loci (Supplementary Fig. 2, Supplementary Data 1). MYC downregulation in EC4-Cas9 cells induced an initial reduction in proliferation but did not affect viability (Supplementary Fig. 3a). Upon MYC downregulation for one week that we considered as the MYC$^{low}$ control, EC4 cells become hepatocyte-like based on the observed enrichment of expression of hepatocyte-specific genes (Supplementary Fig. 3b–e). This result is in line with our previous findings that MYC induces dedifferentiation[14] and that MYC downregulation can induce a differentiation phenotype in HCC[13].

We then performed a genome-wide CRISPR screen in EC4-Cas9 cells with high MYC or low MYC expression using a Mouse GeCKO v2 library[15] that contains 130,209 guide RNAs (gRNAs) targeting 20,611 mouse genes. The library was introduced using a lentiviral infection at low titers so that each cell contained at most one gRNA ("baseline"). Cells were maintained for one week in the presence ("MYC$^{low}$") or absence ("MYC$^{high}$") of doxycycline to allow for the accumulation of genomic mutations by the Cas9/gRNAs and manifestation of knockout-induced phenotypes (Fig. 1a). Then, the frequency of all gRNAs in the different conditions (baseline, MYC$^{low}$ control, MYC$^{high}$ cancer) was assessed by deep sequencing (Supplementary Fig. 4) and MAGeCK-VISPR analysis[16] to determine essentiality scores (Beta scores) for each gene in the genome in the MYC$^{high}$ and the MYC$^{low}$ condition relative to the baseline.

Genes that had a significant (false discovery rate-adjusted *p* value (FDR) < 0.05) negative Beta score were considered essential. We defined genes as a MYC synthetic lethal (MYC-SL) interaction (gene knockouts causing cell death or significant proliferation deficits only in cells with high MYC levels) if the knockout resulted in 1) a negative Beta score indicative of negative selection of gRNAs targeting these genes only in the MYC$^{high}$ condition (FDR < 0.05 and Beta score < 0) and 2) no significant change in cell fitness of the MYC$^{low}$ control cells (FDR > 0.05, or Beta score > 0) (Fig. 1b).

We identified 2395 genes that are required for the survival of HCC MYC$^{high}$ tumor cells, and 682 genes with essential function in MYC$^{low}$ cells. Genes that were essential in both MYC$^{low}$ and MYC$^{high}$ conditions (587 genes) were considered to be required for proliferation and survival irrespective of the level of MYC expression. Gene set enrichment analysis (GSEA) showed that genes, which were previously identified to be essential in human cancer cell lines[17], were preferentially depleted in both MYC$^{high}$ and MYC$^{low}$ conditions while non-essential genes[17] were not (Fig. 1c), illustrating the validity of our screen. Moreover, genes that have been described to be essential in mouse and human[18] were preferentially depleted in both MYC$^{high}$ and MYC$^{low}$ conditions (Fig. 1c)

which is in agreement with previous studies. Therefore, 1808 MYC-SL genes were identified that had essential functions in MYC$^{high}$ but not MYC$^{low}$ tumor cells. (Fig. 1d, (Supplementary Data 2).

Pathway analysis using the Kyoto Encyclopedia of Genes and Genomes (KEGG) revealed that many MYC-SL genes were associated with pathways known to be important for survival in the context of high MYC expression. These include genes involved in ribosomal biogenesis[19–21], RNA transcription[22–24], RNA splicing[25,26], pyrimidine metabolism[27], DNA replication[28], mRNA surveillance, and RNA degradation[29] (Fig. 1d, and Supplementary Data 3). Importantly, genes involved in nuclear to cytoplasmic transport were significantly enriched amongst MYC-SL genes. To our knowledge, this pathway has not been previously implicated in MYC synthetic lethality. Further, there was an overlap between the essential genes identified in our screen with genes previously shown to be required for proliferation and/or survival in a reported screen in human HCC[30,31] (Supplementary Data 4, Supplementary Fig. 5). The set of shared essential genes between both screens contained 861 genes. Of these, 342 genes were essential in human and MYC$^{high}$ murine HCC cells, but not in human pluripotent stem cells (hPSC)[32] and MYC$^{low}$ hepatocyte-like cells. Pathway enrichment analysis of these genes suggested several pathways involving RNA metabolism (ribosome biogenesis, RNA transport, and RNA splicing) to be promising therapeutic targets of MYC$^{high}$ tumors, with few or no treatment effects on healthy tissues (Fig. 1e, Supplementary Data 5).

We also found a correlation of essentiality (Beta scores) in our MYC-driven murine HCC cell line with mean essentiality scores (z-scores) in human HCC cell lines (Fig. 1f). This suggests that the degree a gene product contributes to tumor cell fitness in human HCC is reflected by our murine screening system. Thus, our MYC transgenic mouse model identified essential genes relevant in human HCC.

### CRISPR screen identifies potential tumor suppressor genes

The knockout of antiproliferative genes leads to increased proliferation and clonal expansion of affected cells which is illustrated by a positive Beta score. Genes with antiproliferative function in MYC$^{low}$ cells are likely tumor suppressor genes. We identified 1543 antiproliferative genes only in MYC$^{high}$ but not in MYC$^{low}$ cells, and 173 genes with antiproliferative function in both MYC$^{high}$ and MYC$^{low}$ cells (Fig. 2a). In the MYC$^{low}$ condition, 364 genes had antiproliferative functions (Fig. 2b). These included known tumor suppressor pathway genes[33,34] (Fig. 2c, d, Supplementary Data 2) such as genes involved in p53 signaling (*Trp53*, *Cdkn1a*, and *Cdkn2a*), as well as *Rb1*, *Gata4*, and Hippo signaling pathway components or regulators (*Lats1*, *Nf2*, *Sav2*, *Amotl2*, *Snai2*, *Pals1*, *Vgll4*). Surprisingly, we identified apolipoprotein B (*Apob*) as an antiproliferative gene in both MYC$^{high}$ and MYC$^{low}$ conditions. Combined with the observation that APOB is mutated in about 10% of human HCC and that APOB mutation is associated with decreased survival probability in HCC (Fig. 2e), this finding highlights a tumor suppressive function of APOB in HCC. Our results suggest that this loss of function screen not only identified MYC-SL genes but also known and novel tumor suppressor genes, which might play a role in tumorigenesis.

### CRISPR screen identifies RNA transport as synthetic lethal pathway

We hypothesized that MYC-SL genes, which are most strongly induced by MYC, would be the best therapeutic targets for MYC-driven HCC. Therefore, we performed differential gene expression analysis on primary MYC$^{high}$ and MYC$^{low}$ HCC tumors to identify MYC-driven gene expression changes in situ. We then assessed whether the Beta score of the 1808 identified MYC-SL genes and their MYC-driven gene expression changes correlated.

We found that gene expression stimulated by MYC was associated with MYC-SL gene essentiality in the MYC$^{high}$ condition (Fig. 3a).

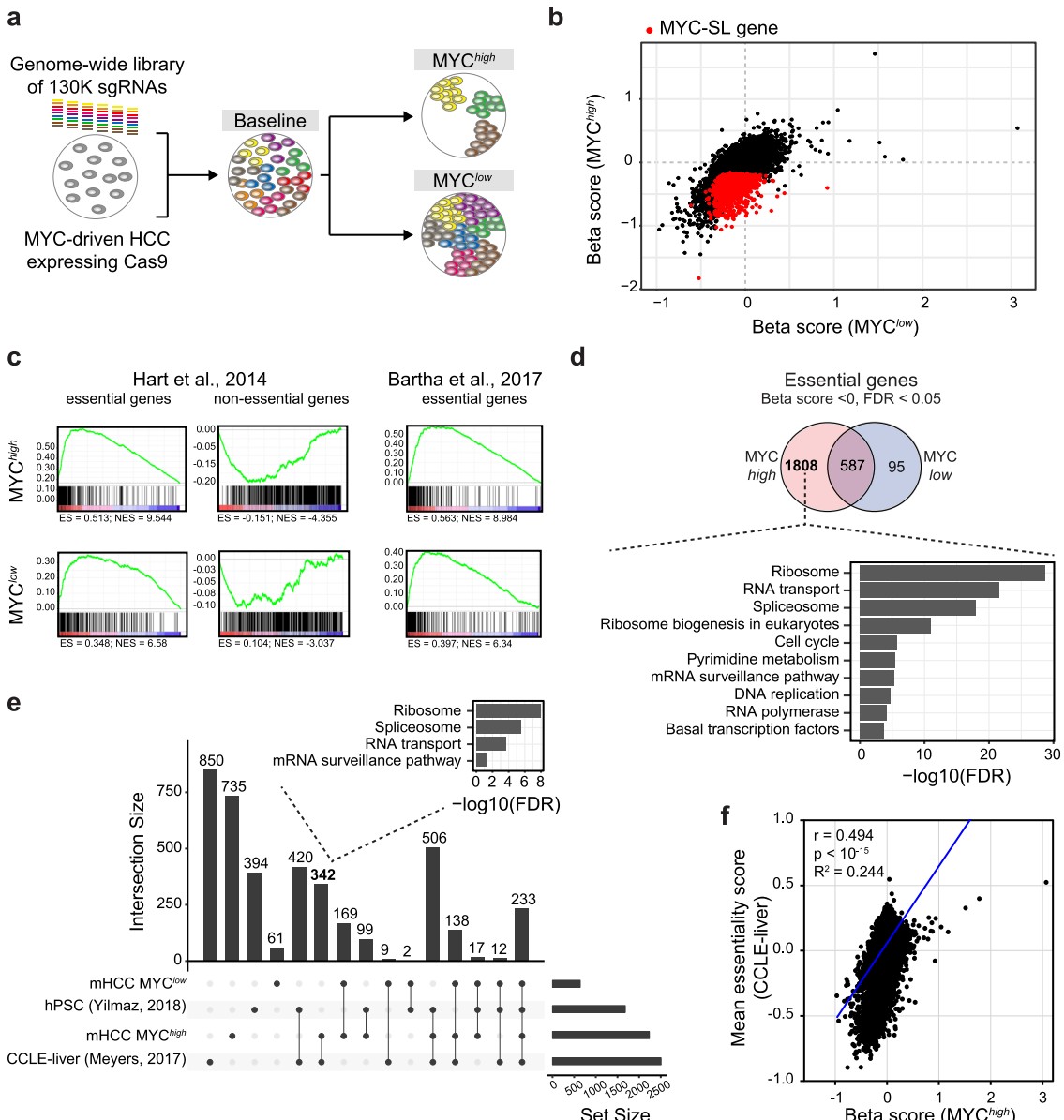

**Fig. 1 | CRISPR-based genome-wide screen identifies essential genes in MYC-driven HCC. a** Design of the CRISPR-based screen to identify vulnerabilities in MYC-driven HCC. A genome-wide sgRNA library (colored bars) is introduced in Cas9-expressing MYC-driven HCC cells to generate a baseline cell pool. Cells are then cultured under MYC$^{high}$ or MYC$^{low}$ conditions resulting in a MYC synthetic lethal (MYC-SL) cell pool and a control cell pool, respectively. Deep sequencing of sgRNAs identifies genes that are essential only when MYC is highly expressed but not essential when MYC transgene expression is shut off. **b** Scatter plot of Beta scores (gene essentiality scores) in the control (MYC$^{low}$; x-axis) versus MYC-SL (MYC$^{high}$; y-axis) condition. Loss of genes highlighted in red significantly reduced fitness of MYC$^{high}$ HCC cells but had no significant effect on MYC$^{low}$ control cells. **c** Gene set enrichment analysis (GSEA) of essential and non-essential genes. Left and middle:

Hart et al. Mol Syst Biol, 2014; Right: Bartha et al. Nat Rev Genet, 2017. **d** Venn diagram of essential genes identified in EC4 cells with high and low MYC levels. KEGG pathway analysis of MYC-SL genes (only essential in MYC$^{high}$ condition) is shown. **e** Comparison of essential gene hits between our genome-wide CRISPR screen (MYC$^{high}$ and MYC$^{low}$ control conditions) and essential genes identified in 13 human HCC cell lines (DepMap, CCLE-liver) and human pluripotent stem cells (hPSC). UpSet plot and numbers of overlapping essential genes between indicated genome-wide CRISPR screens are shown. **f** Correlation analysis of gene essentiality scores from our genome-wide CRISPR screen in MYC-driven murine HCC and essentiality scores in human HCC cell lines (DepMap, CCLE-liver). Pearson's correlation coefficient (r), two-tailed *p*-value, and R² of linear regression (blue line) are shown. For this figure, source data are provided as a Source Data file.

Pathway analysis of the 516 MYC-SL genes that are upregulated by MYC identified 47 RNA transport genes that are involved in RNA metabolism, mRNA surveillance and splicing, and nuclear to cytoplasmic transport of RNA (Fig. 3b, c, Supplementary Fig. 6a, b, Supplementary Data 6–8). These genes are involved in splicing-coupled mRNA/mRNP export like the TREX complex components (*Alyref*, *Thoc1*, *Thoc3*), general mRNA export receptors (*Nxf1*, exportins *Xpo1* and *Xpo5*), and components of the nuclear pore complex itself. This suggests that MYC regulates the expression of specific RNA transport genes required

for nuclear to cytoplasmic transport and also that these genes are essential in MYC-driven tumors.

**Inhibition of XPO1 induces death in MYC$^{high}$ cells**
We further tested if RNA metabolism and transport are potential therapeutic targets in MYC-driven tumors. Among the RNA transport gene products identified, XPO1 and protein arginine N-methyltransferase 5 (PRMT5), have existing selective small molecule inhibitors: XPO1i Selinexor (or KPT-330)[35–37] and PRMT5i

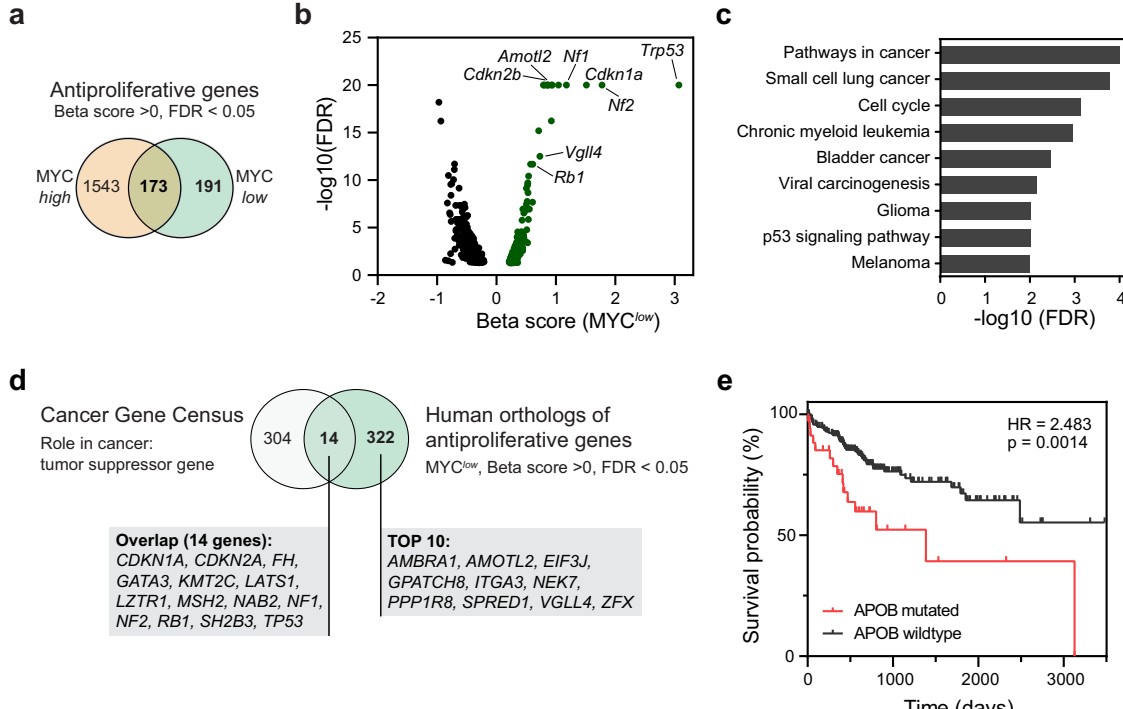

**Fig. 2 | Genome-wide CRISPR screen identifies tumor suppressor genes. a** Venn diagram of genes with antiproliferative function in MYC$^{high}$ and MYC$^{low}$ conditions. **b** Volcano plot of Beta score versus significance (-log$_{10}$(FDR)) of genes with FDR < 0.05 in the MYC$^{low}$ condition. Genes with antiproliferative properties (significant positive Beta score) are highlighted in green. Exemplary tumor suppressor genes are indicated. **c** KEGG pathway analysis of top 50 genes with significant positive Beta score in the MYC$^{low}$ condition. **d** Comparison of known tumor suppressor genes (TSGs) and human orthologues of genes with antiproliferative function in the MYC$^{low}$ condition. Known TSGs were derived from the Cancer Gene Census (https://cancer.sanger.ac.uk/cosmic/census). **e** Survival analysis of TCGA-LIHC assessing the effect of APOB mutation status on survival probability in HCC (*p*-value and hazard ratio (HR) from logrank test). For this figure, source data are provided as a Source Data file.

EPZ15666[38,39]. XPO1 is an export receptor[40–44] that forms a complex with Ran GTPase and drives the transport of multiple classes of RNA and more than a thousand protein targets[45–47]. PRMT5 is an arginine methyltransferase that interacts with MYC[48], regulates RNA metabolism and cellular signaling[49–53] and has been implicated in cancer stemness[54] and aggressiveness[55,56]. MYC-driven HCC cells exhibited decreased ATP levels in response to the inhibition of XPO1 and PRMT5 and fibroblasts were less sensitive towards XPO1 and PRMT5 inhibition than MYC-driven HCC cells (Supplementary Fig. 7a, b). In the human HCC cell line SNU-449, blocking MYC activity using the MAX/MAX homodimer stabilizer MS2-008[57] mitigated the effects of both XPO1 and PRMT5 inhibition on cell metabolic fitness (Supplementary Fig. 7c) and led to a modest increase in biomass accumulation at high inhibitor concentrations compared to control (Supplementary Fig. 7d). Hence, there appears to be a synthetic lethal interaction between MYC and XPO1, and MYC and PRTM5.

We further investigated the potential of XPO1 as a therapeutic target in MYC-driven HCC since XPO1 plays a more direct role in nucleocytoplasmic transport. Xpo1 inhibition suppressed total RNA export in vitro as shown by 5-ethynyl uridine (EU) labeling (Fig. 3d). Xpo1 inhibition increased cell death by 4.4-fold in MYC-driven HCC with high MYC expression levels while hardly affecting survival of cells with low MYC levels (Fig. 3e, Supplementary Fig. 8a, b). Further, XPO1 inhibition decreased cell fitness in other MYC-driven tumors including human P493-6 Burkitt lymphoma-like, a murine MYC-induced T-cell leukemia and a murine *IgH-MYC* B-cell lymphoma cell line (Supplementary Fig. 9). Finally, knockdown of Xpo1 with two specific shRNAs decreased growth of MYC-driven HCC (Fig. 3f, Supplementary Fig. 8c, d). Hence, XPO1 is essential and MYC synthetic lethal in vitro in multiple types of MYC-driven cancer.

## Inhibition of XPO1 induces MYC$^{high}$ HCC regression in vivo
To further explore the role of nucleocytoplasmic transport in MYC-driven cancer, we examined the effect of Xpo1 inhibition by Selinexor in a primary transgenic mouse model of HCC (*LAP-tTA/tet-O-MYC*/FVB/NJ)[13]. We assessed liver tumor volume by magnetic resonance imaging (MRI) before and after treatment (Supplementary Fig. 10). Xpo1 inhibition (3 doses per week for two weeks) resulted in a greater than 95% decrease in tumor volume compared with a 4–12-fold increase in tumor size in vehicle control mice (Fig. 4a–c). In two out of six Selinexor-treated mice, tumors were undetectable by MRI post-treatment. Xpo1 inhibition suppressed tumor proliferation and induced apoptosis as shown by phospho-histone H3 and cleaved caspase-3 staining, respectively (Fig. 4d–f). After short-term treatment, the histology of the remaining tumor cells appears both necrotic and with decreased nuclear to cytoplasmic ratio (Fig. 4d, Supplementary Fig. 11), further corroborating the effect of Xpo1 inhibition on tumor fitness. In contrast, Xpo1 inhibition did not induce changes in proliferation or apoptosis in adjacent normal liver tissue (Supplementary Fig. 11). Thus, short-term therapeutic inhibition of the nucleocytoplasmic transport gene, Xpo1, induces tumor regression without affecting normal adjacent liver in an autochthonous transgenic mouse model of MYC-induced HCC.

## MYC-SL nucleocytoplasmic transport genes predict poor survival, and XPO1 inhibition suppresses HCC PDX growth
We examined the role of nuclear to cytoplasmic transport genes including XPO1 in human cancer. First, we examined if the expression of MYC-SL nuclear to cytoplasmic transport genes predicted prognosis of human HCC patients. The increased expression of MYC-SL nuclear to cytoplasmic transport genes predicted poor clinical outcome in human HCC patients (LIHC: human HCC)[58]. The prognostic power

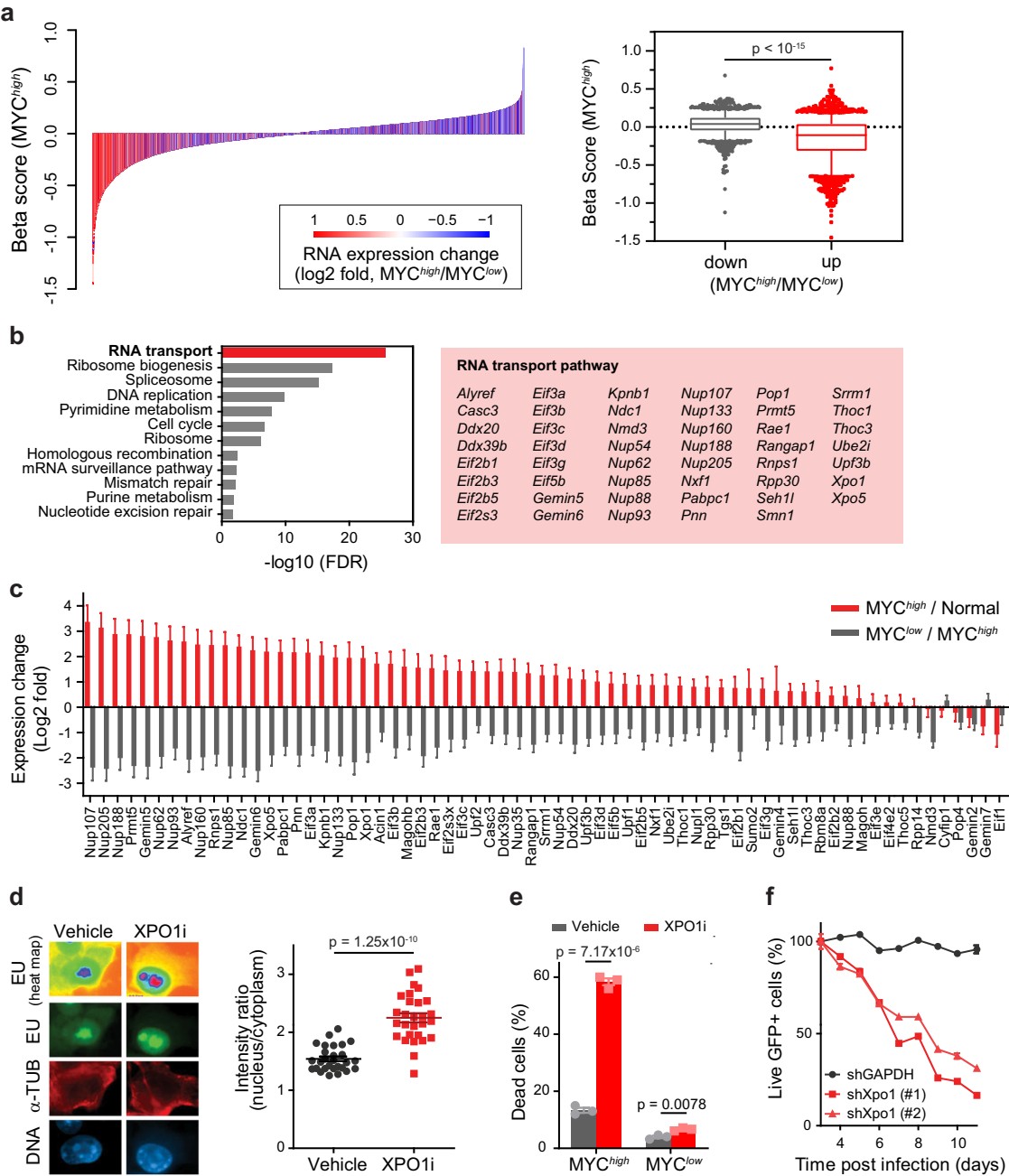

**Fig. 3 | MYC-driven HCC cells depend on XPO1 activity. a** Correlation analysis of MYC-regulated gene expression and gene essentiality. *Left:* Differential gene expression (MYC$^{high}$/MYC$^{low}$, log2-fold change) is plotted against gene essentiality. Gene expression was determined in the context of MYC-induced HCC development (MYC$^{high}$ HCC ($n = 6$ mice)) and MYC inactivation-induced tumor regression (MYC$^{low}$ HCC ($n = 3$ mice)) by RNA sequencing. *Right:* Gene essentiality (Beta score) is plotted for genes that are either significantly up or down regulated in MYC$^{high}$ compared to MYC$^{low}$ HCC (adjusted *p*-value < 0.05). Box: 25–75 percentile, whiskers: 5–95 percentile, line: median, data points outside of 5–95 percentile are shown. The *p*-value was determined by two-tailed Mann-Whitney U test. **b** Results of KEGG pathway analysis of MYC-upregulated MYC-SL genes. Significantly enriched pathways (FDR < 0.05) are shown. RNA transport pathway genes identified as MYC-SL are highlighted in red box. **c** Gene expression changes of MYC-SL RNA transport genes in the context of MYC-induced HCC development (red: MYC$^{high}$ HCC ($n = 6$ mice) vs. normal liver tissue ($n = 2$ mice)) and MYC inactivation-induced tumor regression (grey: MYC$^{low}$ HCC ($n = 3$ mice) vs. MYC$^{high}$ HCC). Gene expression was determined by RNA sequencing (bars: mean, error bars: SE). **d** Epifluorescence images of total RNA (heat map, and green) by EU labeling in MYC-driven HCC cells upon XPO1 inhibition by Selinexor (scale bar = 5 μm). Staining of alpha-tubulin (red) and DNA (blue) were used for segmentation of nucleus and cytoplasm. Quantification of RNA abundance in the nucleus versus cytoplasm per cell ($n = 28$ cells examined in one experiment). Lines are means +/− SEM. The *p*-value for the two-tailed Student's t-test is shown. **e** Flow cytometric analysis of EC4 cells with high MYC expression (MYC$^{high}$) or MYC transgene expression shut off (MYC$^{low}$) by doxycycline treated with XPO1 inhibitor Selinexor. Survival was assessed by propidium iodide/AnnexinV staining. Graph shows quantification of three independent experiments (bars: mean, error bars: SEM). The *p*-value for the two-tailed Student's t-test is shown. **f** Flow cytometric analysis of MYC-driven HCC cells expressing shRNAs targeting Xpo1 (two different Xpo1-targeting shRNAs) or GAPDH as a control in combination with GFP as a reporter. Live cells were detected by propidium iodide exclusion. Shown is the percentage of shRNA expressing (GFP+) live cells over time. Lines connect means of technical triplicates. For this figure, source data are provided as a Source Data file.

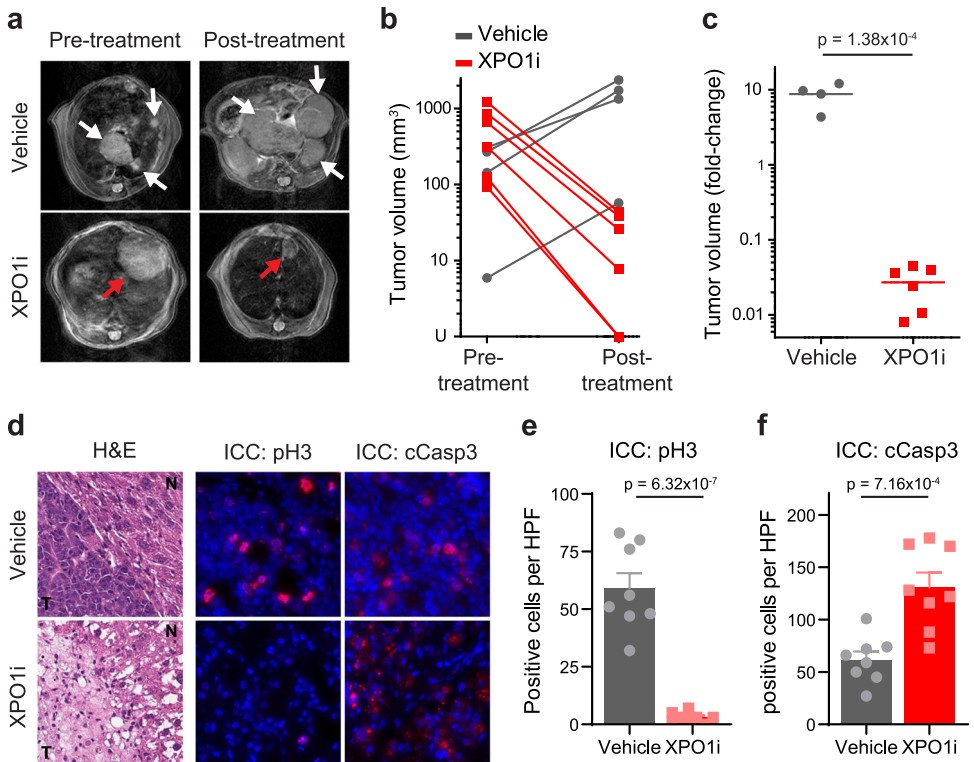

**Fig. 4 | XPO1 inhibition induces tumor regression in primary mouse model of MYC-induced HCC. a** Representative MRI scans of liver tumors in *LAP-tTA/tet-O-MYC*/FVB/N mice treated with vehicle or Selinexor (3 doses per week for two weeks). The tumor nodules are indicated by white or red arrows. **b, c** Absolute and relative fold changes in liver tumor volume in *LAP-tTA/tet-O-MYC*/FVB/N mice treated with vehicle (*n* = 4 mice) or Selinexor (*n* = 6 mice). In (**c**), lines represent means which were compared using two-tailed Student's t-test. **d–f** Immunofluorescence staining and quantification of phospho-histone H3 and cleaved-caspase 3 in liver tumors of mice after short-term treatment with vehicle or Selinexor (3 doses per week, one week). Cell nuclei were counterstained with 4′,6-diamidino-2-phenylindole (DAPI). The histology of tumors (T) and adjacent normal liver tissues (N) are shown by hematoxylin and eosin staining. Scale bars = 50 µm. Samples were collected from at least three mice in each group. Bars in (**e**, **f**) represent means of *n* = 8 high-power fields (HPFs) of one representative biological sample, error bars represent SEM, *p*-values were determined using two-tailed Student's t-test. For this figure, source data are provided as a Source Data file.

(z-score) of MYC-SL RNA transport genes is significantly associated with poor prognosis in human HCC compared to the reference distribution of z-scores (Fig. 5a). To examine whether HCC patients with high and low MYC activity have different prognoses depending on XPO1 expression, we stratified the TCGA-LIHC cohort by expression of MYC-regulated hallmark genes and evaluated the effect of XPO1 expression on survival probability in these groups. Overexpression of MYC hallmark genes in human HCC patients predicted decreased overall survival (HR = 2.0, 95% CI 1.3–3.3) (Fig. 5b). In patients with high MYC activity (MYC_Hallmarks_V2_score*high*), XPO1 had prognostic power and high XPO1 expression predicted poor outcome (HR = 4.1, 95% CI 1.6–10.7). However, XPO1 expression had no prognostic power in patients with low MYC activity (MYC_Hallmarks_V2_score*low*) (HR = 1.4, 95% CI 0.4–4.5) (Fig. 5c). Therefore, MYC-SL RNA transport genes and XPO1 specifically predict poor prognosis, and XPO1 is a prognostic biomarker only in HCC with high MYC activity.

Second, we examined if XPO1 inhibition influenced the in vivo growth of human HCC patient-derived xenografts (PDX). We found that XPO1 inhibition reduced the growth of HCC PDX. This effect was more pronounced in a PDX with high MYC expression as compared to a PDX with low MYC levels (Fig. 5d, e) and increased apoptosis of tumor cells as measured by cleaved caspase 3 (cCasp3) (Supplementary Fig. 12). Hence, inhibition of the nuclear and cytoplasmic transport gene, XPO1, inhibits the growth of human HCC PDX.

## Discussion

We have performed a comprehensive assessment of MYC synthetic lethal (MYC-SL) interactions identifying 1808 genes that are only essential in MYC*high* HCC cells but not in MYC*low* normal hepatocyte-like cells. Our MYC-specific CRISPR/Cas9-based genome-wide screen identified nuclear to cytoplasmic transport as a novel MYC-SL pathway. We found that 71 out of 170 genes in the nuclear to cytoplasmic transport of RNA and/or proteins were MYC synthetic lethal. In general, the increased expression of these nuclear to cytoplasmic transport genes predicted poor prognosis in human HCC. The majority were co-expressed with MYC in our preclinical HCC model. As a proof-of-principle, we targeted one of the nuclear to cytoplasmic transport genes, XPO1, demonstrating its inhibition induced tumor cell death both in vitro and in vivo, in mouse and human MYC-driven cancer cell lines, in a transgenic mouse model of MYC-driven HCC, and in human HCC PDX. XPO1 as a potential therapeutic target specifically for MYC-driven cancers has not been previously recognized. Further, our observations also provide comprehensive insights into specific genes and pathways that confer cellular fitness in MYC-driven cancers.

Our rationally designed genome-wide CRISPR screen using a conditional MYC expression system uniquely allowed us to determine genes that are only essential to cancer cells with high, and not low, MYC expression. This enabled us to determine many previously unknown MYC-SL genes and genes with potential tumor suppressive properties. The robustness of our screen is validated by the finding that we did identify several known MYC-SL genes including *Brd4*[59−63], *Atr*[64], *Sae1*[60], *Prmt5*[26], and *Birc5*[65]. Moreover, by using a conditional MYC system, we demonstrate that some of the genes previously identified as MYC-SL, like *Mtor*[66], *Cdk9*[67], or *Wdr5*[68], may not be specific for MYC-driven cancers. While these genes may still be effective targets for cancer treatment, they may not necessarily be MYC-SL.

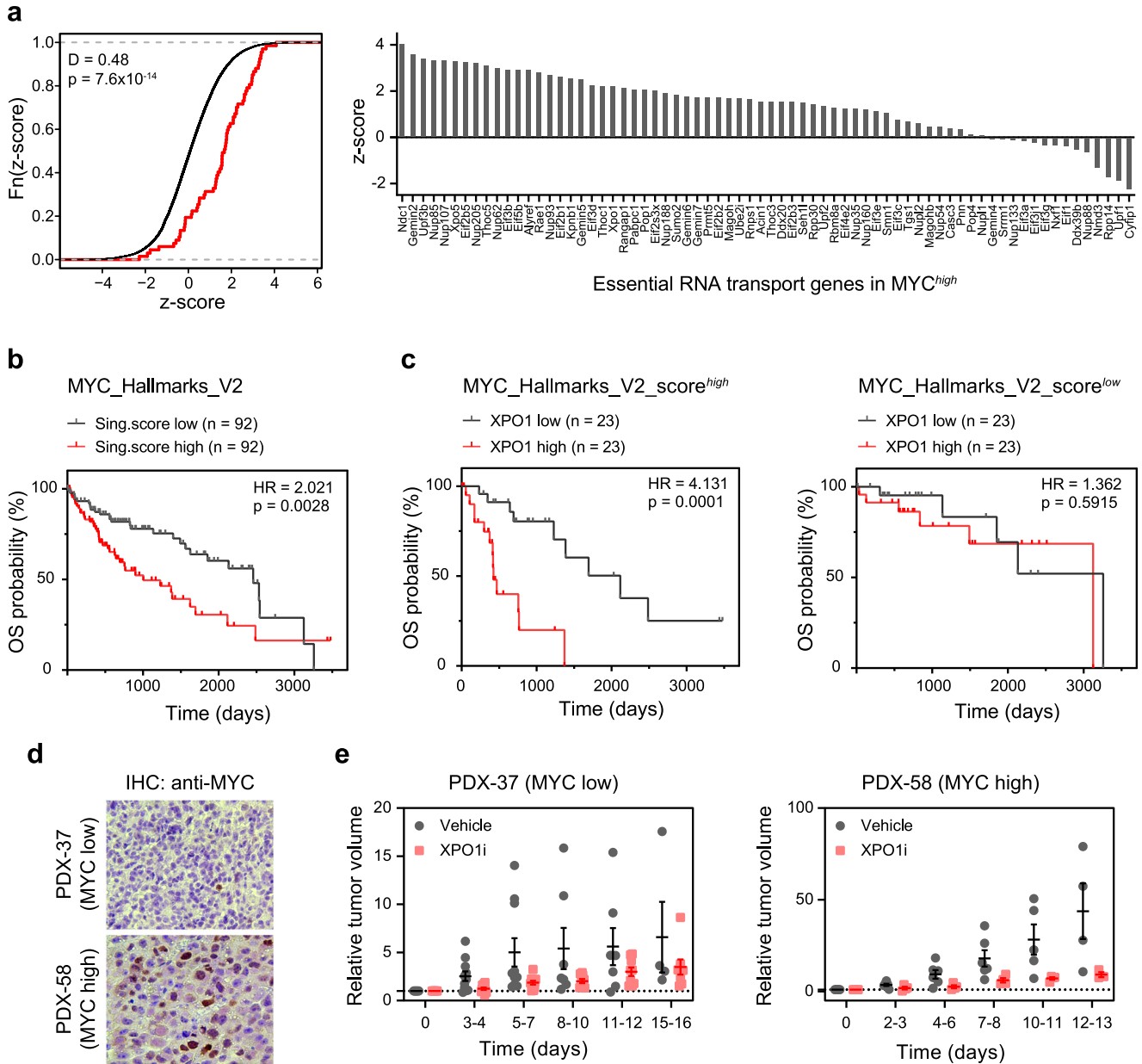

**Fig. 5 | Increased expression of MYC-SL nucleocytoplasmic transport genes is associated with poor prognosis in human HCC and XPO1 inhibition reduces growth of patient-derived xenografts. a** Association of MYC-SL RNA transport genes with disease outcome in human HCC. Survival z-scores were calculated based on Cox Regression using TCGA data on liver hepatocellular carcinoma (LIHC RNA-seq and clinical data). Positive z-scores are associated with poor prognosis. Left: Z-scores of MYC-SL RNA transport genes (red) are compared to the genome-wide random distribution (black). Right: Prognostic association of individual MYC-SL RNA transport genes in human HCC. **b** Survival analysis of TCGA LIHC data subdivided by expression of MYC hallmark genes (HALLMARK_MYC_TARGETS_V2, M5928) assessed by singscore[92]. Graphs show upper and lower quartiles (*p*-value and HR from two-sided logrank test). **c** Survival analysis of subgroups with high and low MYC activity shown in (**b**) based on XPO1 RNA expression level (*p*-value and HR from two-sided logrank test). **d** Representative images of MYC expression in two biologically independent HCC PDX models assessed by immunohistochemistry. At least three high power field images were aquired per PDX. Scale bar = 20 µm. **e** Relative tumor volume of HCC patient-derived xenografts (PDX) with high or low MYC expression in NSG mice treated with vehicle (grey symbols) or XPO1 inhibitor Selinexor (red symbols). PDX samples obtained from two HCC patients (PDX-37, and PDX-58) were examined over two independent experiments (PDX-37: vehicle (*n* = 11 mice), XPO1i (*n* = 11 mice); PDX-58: vehicle (*n* = 6 mice), XPO1i (*n* = 4 mice). Lines are means +/− SEM. For this figure, source data are provided as a Source Data file.

Our screen also uncovered tumor suppressor genes whose loss cooperates with MYC to promote cancer. Examples for identified well-known tumor suppressors include genes such as *Trp53*, *Rb1*, *Nf1* and *Nf2*, and Hippo signaling pathway components. However, other genes with similar tumor suppressive function in our screen, such as *Ambra1*, *Gpatch8*, and *Spred1*, may be less appreciated as tumor suppressors. Thus, our genome wide CRISPR-screen served as a powerful approach to determine specific MYC-SL

genes and pathways as well as genes with anti-tumor suppressive function.

Our report identified nuclear to cytoplasmic transport, a key step between transcription/splicing and translation, as a MYC-regulated and MYC synthetic lethal pathway. While MYC is known to regulate several aspects of RNA metabolism including ribosome biogenesis, RNA transcription, splicing and translation, its central role in regulating and depending upon efficient nuclear to cytoplasmic transport has

not been reported before. Of the 170 genes known to be involved in nuclear to cytoplasmic transport, we found that 71 were individually MYC-SL, and 47 were co-expressed with MYC, thus underscoring the robustness of dependency of MYC on nuclear export to drive cancer. One of the nuclear exporters identified as MYC-SL in our screen was Xpo1, which is involved in the transport of thousands of proteins, ribosomal RNAs and less abundant RNA species[69]. High XPO1 expression has been associated with poor prognosis in multiple cancers including ovarian cancer[70], osteosarcoma[71], acute myeloid leukemia[72], pancreatic adenocarcinoma[73], and neuroblastoma[74]. We show here that inhibition of XPO1-mediated nucleocytoplasmic transport blocked growth in vitro, induced tumor regression in vivo in a transgenic mouse, and inhibited human HCC PDX, more effectively in MYC*high* cancers. Apart from XPO1, our screen also identified other proteins transported by XPO1 to be products of MYC-SL genes. Interestingly, several of these proteins are also known to be important in MYC-driven cancers like ribosome components (such as *Rps8*), multiple subunits of the eukaryotic translation initiation factor 3 complex, and proteins important for MYC-induced autophagy (such as *Atg3*). Taken together, these findings highlight the critical role of XPO1-mediated nuclear to cytoplasmic transport in promoting MYC-driven tumor growth.

MYC is a master transcription regulator, and export of RNA species from the nucleus to the cytoplasm is a crucial step for efficient gene expression. We show here that several genes involved in mRNA processing and transport play essential roles in MYC-driven cancer cells. These included components of the exon junction complex (such as *Ayref*, *Pinin*, *Rnps1*, and *Ddx39b*), transcription export complex (such as *Alyref*, *Thoc1*, and *Thoc3*), and the major mRNA exporter *Nxf1*. We note that these findings are in accordance with a recent study showing that XPO1 transports THOC4-bound mRNAs that encode for DNA damage repair proteins[75], which in turn help protect cancer cells from MYC-induced DNA damage[76,77]. Another MYC-SL gene we identified was *Nmd3*, which encodes a 60S ribosomal export protein which together with Xpo1, transports ribosomal subunits from the nucleus to the cytoplasm[78–80]. The finding that multiple genes involved in RNA transport, by a XPO1-independent or -dependent mechanism, have synthetic lethal interactions with MYC underscores the essentiality of this pathway for MYC function. Thus, altered nuclear export of RNAs is a dependency of MYC-driven cancers.

MYC is a key oncogenic protein in human cancer but remains a difficult protein to target directly. Thus, the MYC-SL genes we have identified serve as valuable resource to determine strategies to indirectly, but specifically, target MYC in cancer. Importantly, we anticipate that targeting a combination of MYC-SL gene products may lead to synergistic drug interactions. It has recently been reported that XPO1 inhibition synergizes with the inhibition of other MYC-SL gene products we identified in our screen, such as DNMT1[81,82], RRM1/2[83], or BRD4[84]. Based on the synthetic lethal interaction of MYC and XPO1 described here, we demonstrate that XPO1 inhibition was most effective in tumors with high MYC activity. In line with this, in human HCC, XPO1 levels were predictive of clinical outcomes only in tumors with high, but not low, expression of MYC activation signature. Selinexor is a highly selective XPO1 inhibitor which has been FDA approved for the treatment of refractory multiple myeloma and diffuse large B-cell lymphoma, and is being currently evaluated in clinical trials for other cancers (for example, currently in Phase 3 or 4: NCT05028348, NCT05611931, NCT04562389, NCT05726110). However, a biomarker predicting response to XPO1 inhibition is lacking. Our study demonstrates that stratification of tumors based on MYC activation status will likely enable more precise selection of patients for RNA transport inhibitors. More importantly, we identify several MYC-SL genes in the RNA transport pathway, apart from XPO1, which are yet to be targeted and can serve as rational, and specific targets against MYC-driven cancers.

Our results suggest that MYC may globally regulate nuclear to cytoplasmic transport. This may be a mechanism by which MYC maintains tumorigenesis. We identified multiple nuclear to cytoplasmic transport genes to have synthetic lethal interactions with MYC, thus highlighting the essential nature of this pathway. MYC may control a multitude of cellular functions through regulation of nuclear transport of proteins and RNA, thus promoting tumor progression. Using a MYC-conditional murine in vitro system, we have comprehensively identified 1808 cell-intrinsic MYC-SL interactions. In our primary murine HCC model, XPO1 inhibition induced remarkable tumor regression. Amongst the advantages of our model is that it is fully immune competent. Drug efficacies could be underestimated in immunocompromised xenograft-based models which lack the contribution of an anti-cancer immune response. For instance, inhibiting a MYC-SL target may also have beneficial on-target but off-tumor effects on the tumor microenvironment. A recent example is the stimulation of anti-tumor immunity by BRD4 inhibition[85–87]. Given that MYC regulates the tumor microenvironment[88–93], in future studies, we will evaluate MYC synthetic lethal interactions in hosts with an intact immune system. Here, we performed a MYC-specific genome-wide CRISPR synthetic-lethality screen and identify nuclear to cytoplasmic transport as a therapeutic vulnerability of MYC-driven cancers.

## Methods

All procedures and methods were conducted in accordance with federal and state regulations as well as Stanford University's institutional guidelines and policies. All work involving materials classified as biosafety level 2 or higher has been approved by Stanford University's Administrative Panel on Biosafety. Animal experiments described in this study have been approved by Administrative Panel on Laboratory Animal Care at Stanford University (protocols APLAC-14045, and APLAC-10563) and comply with all federal and state regulations governing the humane care and use of laboratory animals, including the USDA Animal Welfare Act, and Stanford's Assurance of Compliance with PHS Policy on Humane Care and Use of Laboratory Animals. Tissue sample collection from HCC patients following informed consent was approved by the Stanford University Institutional Review Board (IRB-28374).

### CRISPR library screening and data analysis

The CRISPR library screening was performed following published protocols[94]. The pooled mouse GeCKO v2 CRISPR library[15] containing 130209 gRNAs was obtained from Addgene. Briefly, the library was amplified by electroporation into Endura competent cells (Lucigen). After lentiviral packaging of the library, EC4 cells with stable Cas9 expression were infected at low viral titers so that approximately 10% of the cells were infected. The infected cells were treated with puromycin (1 μg/ml) for two days before screening. These antibiotic-selected cells served as the baseline pool and were further passaged and separated into two conditions: the MYC*high* (cancer) pool that continued to overexpress MYC and the MYC*low* (control) pool that was treated with doxycycline to shut off MYC expression.

The gRNA libraries in the screened populations were subsequently isolated by PCR amplification and characterized by hi-seq. The CRISPR/Cas9 screening data was processed and analyzed using MAGeCK-VISPR[16]. For two conditions (SL and control), Beta scores, which provide a quantification of gene essentiality, were generated from MAGeCK-VISPR's maximum likelihood estimation algorithm. Additionally, $p$-values were determined by the Wald test and FDR-adjusted $p$-values were given by the Benjamini-Hochberg method. Statistical significance required FDR < 0.05. To identify genes synthetic lethal to high MYC but not low MYC levels, we used the following criteria: Beta score < 0 and statistical significance (SL.FDR < 0.05) in the SL condition (indicating negative selection) but no statistically significant selection in the control condition (CONTROL.FDR > 0.05).

The analysis and plotting of data were performed using the R statistical programming language. KEGG pathway enrichment analysis was performed using DAVID[95,96].

## Cell lines and cell culture

Conditional MYC-driven murine HCC cells, EC4 cells, were derived from a liver cancer generated in a *LAP-tTA/tet-O-MYC*/FVB/N mouse. Murine lymphoma cells were derived from *EµSRα-tTA/tet-O-MYC*/FVB/N or *Eµ-MYC* mice. SNU449, BJ5-tA, and NIH3T3 cells were obtained from ATCC (CRL-2234, CRL-4001, and CRL-1658, respectively; characterization by STR profiling performed by cell bank). 293FT cells were purchased from Thermo Fisher Scientific. EC4, SNU449, BJ5-tA, 293FT and NIH3T3 cells were maintained in Dulbecco's Modified Eagle's Medium (DMEM, Gibco) with 4.5 g/l D-Glucose, supplemented with 10% fetal bovine serum, 2 mM L-glutamine (Gibco), 1× MEM Non-Essential Amino Acids (Gibco), 1 mM sodium pyruvate, 100 IU/ml penicillin, and 100 µg/ml streptomycin. The human lymphoma-like cell line P493-6[97,98] was kindly provided by Prof. Chi V. Dang (Ludwig Institute for Cancer Research, Johns Hopkins University School of Medicine, Baltimore, USA), and maintained in Roswell Park Memorial Institute 1640 medium (RPMI, Gibco) with GlutaMAX containing 10% fetal bovine serum, 100 IU/ml penicillin, 100 µg/ml streptomycin. Murine lymphoma cells were maintained in Roswell Park Memorial Institute 1640 medium (RPMI, Gibco) with GlutaMAX containing 10% fetal bovine serum, 100 IU/ml penicillin, 100 µg/ml streptomycin, and 50 µM 2-mercaptoethanol. MYC-driven murine cancer cell lines are available from corresponding author upon request. Cells were cultured in a humidified 5% $CO_2$ atmosphere at 37 °C. All cell lines were regularly confirmed to be mycoplasma-free using PCR- or mycoplasmal enzyme-based methods.

## Gene functional enrichment analysis

Gene Set Enrichment Analysis was performed using the GSEA (v4.1.0) software from the Broad Institute (https://www.gsea-msigdb.org/gsea/index.jsp). Rank scores for gene essentiality were calculated as $\log_{10}$(FDR) multiplied by the sign of the Beta score, such that essential (negatively selected) genes receive a higher rank score than non-essential genes. The GSEAPreranked algorithm was used on this ranked gene list. To assess gene set enrichment in RNA expression data sets, normalized counts (obtained using DESeq2, see "RNA isolation and RNA-seq data analysis") were used. The Database for Annotation, Visualization and Integrated Discovery (DAVID, https://david.ncifcrf.gov/) was used for gene enrichment and functional annotation analysis (one-sided Fisher's exact test).

## EU-labeling and visualization of total RNA

EC4 cells were labeled with 1 mM 5-ethynyl uridine (EU) for 5 h in the presence of 150 nM Selinexor or vehicle (DMSO control) at 37 °C and 5% $CO_2$. Cells were then fixed with 4% formaldehyde in PBS at room temperature for 20 min, washed consecutively with PBS, 50 mM $NH_4Cl$/PBS, PBS (10 min each) and permeabilized with 0.2% Triton-X100/PBS. After washing with PBS, EU was labeled with Alexa-647-azide using the Click-IT imaging kit (Invitrogen) following the manufacturer's instructions. Images were acquired on a DMI 6000 B (Leica) epifluorescence microscope. Mean fluorescence intensities in regions of interest in the cytoplasm and in the nucleus omitting nucleoli were determined using ImageJ (v1.47k). Cytoplasmic to nuclear intensity ratios in Selinexor and DMSO-treated cells were compared by two-tailed Student's t-test.

## RNA isolation and RNA-seq data analysis

RNA sequencing of RNA isolated from primary tumor tissue ($MYC^{high}$, $MYC^{low}$ for one week) isolated from LAP-tTA/*tet-O*-MYC mice was performed by BGI, RNA sequencing of RNA isolated from EC4 cells was performed by Novogene. STAR (v2.5.4b) two-pass alignment was used

to align RNA-seq raw reads to the GRCm38 (mm10) mouse primary genome assembly (gene annotation version M15), which was modified to include the human MYC sequence. DESeq2 (v1.22.2) was used for differential gene expression analysis (two-sided Wald test was used for statistical comparison).

TCGA RNA-seq count data was retrieved from the National Cancer Institute's Genomic Data Commons (GDC) data portal. DESeq2 (v1.22.2) was used to normalize RNA-seq read counts and ComBat was used for batch effect correction. Survival data was queried from TCGA using the R library, TCGAbiolinks. For each gene, a z-score was derived from the Cox proportional hazards regression model fitting gene expression to survival.

## Flow cytometric analysis of cell viability

For apoptosis analysis, cells were stained with 7-AAD and PE-Annexin-V (Becton Dickinson) and analyzed on a FACScan flow cytometer following manufacturer's instructions. FACS data was analyzed with FlowJo (v10.8.1) software (Tree Star). The apoptotic cell populations were defined by positive staining of both Annexin-V and 7-AAD. For shRNA-mediated knockdown of Xpo1, lentiviral particle were produced by co-transfection of 293FT cells with pMD2.G, psPAX2, and the indicated pGIPZ-shRNA construct. psPAX2 (Addgene plasmid # 12260; http://n2t.net/addgene:12260) and pMD2.G (Addgene plasmid # 12259; http://n2t.net/addgene:12259) were gifts from Didier Trono. pGIPZ lentiviral constructs were obtained from Horizon Discovery: shXpo1 #1 (clone V3lMM_475788, mature antisense sequence: ACTTCTCCAACTTGAACCT, shXpo1 #2 (clone V3LMM_475791, mature antisense sequence: TGAACTGTCTGGTCTGTCT). pGIPZ GAPDH lentiviral shRNA construct (Horizon Discovery) was used as control. Two days after co-transfection of 293FT cells with lentiviral constructs, lentiviral particles were concentrated using Lenti-X Concentrator (Takara) and directly used to transduce EC4 cell by spininfection in the presence of polybrene. Cell viability of shRNA-expressing, GFP-positive cells was assessed by propidium iodide staining followed by flow cytometry (BD Accuri C6 Plus).

## Western blot and immunohistochemical analysis

The following antibodies were used for Western blot (WB) anlysis: Anti-MYC clone 9E10 (1:5000, Millipore-Sigma, M4439), Anti-MYC clone Y69 (1:1000, Abcam, Ab32072), Anti-Exportin-1 clone D6V7N (1:1000, Cell Signaling Technology, 46249), Anti-GAPDH clone 14C10 (1:5000, Cell Signaling Technology, 2118), Anti-alpha Tubulin clone DM1A (1:2000, Millipore-Sigma, T9026), Anti-beta Actin (1:1000, Cell Signaling Technology, 4967), Anti-rabbit IgG IRDye800CW (1:10000, Licor, 926-32211), Anti-mouse IgG IRDye680RD (1:10000, Licor, 926-68070), Anti-rabbit IgG-AP (1:5000, Invitrogen, G-21079), Anti-mouse IgG/IgM-AP (1:5000, Invitrogen, 31330). The following antibodies were used for immunohistochemistry (IHC): Anti-MYC clone EP121 (1:150, Millipore-Sigma, 395 R), Anti-cleaved Caspase-3 (Asp175) (1:100, Cell Signaling Technology, 9661), Biotinylated anti-rabbit IgG (1:500, Vector Laboratories, BA-1000-1.5). For immunofluorescence (IF) stainings, we used the following antibodies: Anti-phospho Histone-3 (Ser10) (1:200, Cell Signaling Technology, 9701), Anti-cleaved Caspase-3 (Asp175) (1:100, Cell Signaling Technology, 9661), Anti-rabbit IgG AlexaFluor568 (1:400, Invitrogen, A-11011). Immunofluorescence or bright field images were taken with 20x objectives on a Leica DMI6000 B microscope and quantified using MetaMorph (v7.8) image analysis software.

## Cell viability assays

Cells were seeded at a density of 1000–2000 cells/well in 96-well plates. Cells were treated with different concentrations of Selinexor (from 0.33 nM to 10 µM) for 72 h. MYC expression was shut off by treating the cells with doxycycline (100 ng/ml), or MYC was inactivated by treatment with MS2-008 (10 µM) 24 h prior addition of other

inhibitors. All experiments were performed in quadruplicate. Cell viability was estimated using the CellTiter-Glo Luminescent Cell Viability Assay (Promega) or Sulforhodamine B Assay (Abcam). The log-transformed concentration values and the normalized luminance or absorbance data were fitted to a four-parameter logistic equation using Prism (v3.9.1) software (GraphPad).

### In vivo efficacy studies

All experiments using patient material or research animals were performed according to the ethical approval provided by the IRB and APLAC boards of our institution. Mice were generally maintained on a 12 hours-dark/ 12 hours-light cycle at 70 °F (+/−2 °F) and 50% (+/−20%) humidity. The generation of *LAP-tTA*/*tet-O-MYC*/FVB/N mice has been described previously[18]. The breeder lines, *LAP-tTA*/FVB/N and *tet-O-MYC*(36)/FVB/N (Felsher laboratory), were crossed in the presence of doxycycline to generate *LAP-tTA*/*tet-O-MYC*/FVB/N mice. Only male mice of this strain carry the tet-O-*MYC* transgene and develop liver cancer upon doxycycline withdrawal. To induce MYC-driven HCC development, doxycycline was withdrawn when mice are 21 days old and weaned. Mice were imaged regularly by MRI (7T, Bruker, Agilent conversion) to detect developing tumors. Mice with tumors of 50–200 mm$^3$ volume were randomly assigned to vehicle or Selinexor treatment groups. Selinexor (Selleckchem) was dissolved in vehicle (1% Pluronic F-68 and 1% PVP-K29/32, 25 mg/kg). Both Selinexor and vehicle were administered by oral gavage to tumor-bearing mice three times a week for 6 doses total. Tumor volume pre- and post-treatment was calculated based on the T2 weighted image stacks using the Osirix Lite (v12.5.0) software. For short-term treatment, the mice were treated with either vehicle or Selinexor for three doses and sacrificed the following day after the third dose.

Patients who underwent hepatic resection for hepatocellular carcinoma were identified from chart review. Informed consent was obtained from all patients prior to obtaining tissue sample. Freshly resected tissue of HCC was chopped, mixed with an equal volume 0.1 ml of Matrigel, and implanted subcutaneously into bilateral flanks of 6–8 week old NOD/SCID/IL2Rγ-/- (NSG) mice. Male and female NSG mice were used and distributed equally between groups. NSG breeder mice were obtained from Jackson Laboratories, experimental animals were generated at Stanford University. The mice were euthanized, the tumor tissue retrieved and serially passaged until F3 generation. Histopathologic assessment was used to confirm that tumor had not drifted significantly from original human tumor. The F3 generations of two independent patient samples (PDX-37, and PDX-58) were used for the XPO1 inhibitor efficacy study. When F3 tumors were established and palpable, mice were treated by oral gavage with vehicle (1% Pluronic F-68, 1% PVP-K29/32) or with 30 mg/kg Selinexor dissolved in vehicle. Caliper measurements were performed daily and tumor volume was estimated according to the formula Volume=1/2(L*W$^2$), where L and W are the length and width of the tumor, respectively. A maximal tumor size of 1.70 cm (largest diameter) was permitted by Stanford University's Administrative Panel on Laboratory Animal Care (APLAC) and it was not exceeded. Humane endpoints for early euthanasia included signs of pain, labored breathing, or the inability to move and stand normally. Other early euthanasia criteria were a total body weight loss greater than 20%, or more than 10% body weight loss in a single week. Mice were euthanized by gradual carbon dioxide displacement and subsequent confirmatory cervical dislocation.

### Analysis of TCGA RNA-sequencing and survival data

Analysis of human HCC RNA-sequencing and clinical data was performed as described previously[14]. Survival data was obtained from NIH Genomic Data commons (https://gdc.cancer.gov/about-data/publications/pancanatlas). Stratification of the TCGA-LIHC cohort by expression of MYC-regulated hallmark genes was performed using the gene set 'HALLMARK_MYC_TARGETS_V2' (available at https://www.gsea-msigdb.org/gsea/msigdb/index.jsp) and the R package singscore (v1.14.0)[99,100].

### Statistics and reproducibility

Statistical tests used for data analysis are indicated in the figure legends or in the Materials and Methods section. No statistical method was used to predetermine sample size. No data were excluded from the analyses. If not stated otherwise, the experiments were not randomized and the investigators were not blinded to allocation during experiments and outcome assessment. For in vivo studies of XPO1 inhibitor activity, HCC bearing mice were randomly assigned to vehicle and inhibitor treatment groups. Measurements of PDX tumor sizes were performed by a blinded investigator.

### Reporting summary

Further information on research design is available in the Nature Portfolio Reporting Summary linked to this article.

## Data availability

All raw sequencing data generated in this study have been deposited in Gene Expression Omnibus and are available under accession code Series GSE205132. Sub-series GSE205130 contains RNA sequencing data, sub-series GSE205131 contains genome-wide CRISPR/Cas9 screening data. Processed sequencing data, quantification of imaging and survival data are provided in the Supplementary Information or the Source Data. Normalized human RNA-seq quantification data from The Cancer Genome Atlas (TCGA) is available via the UCSC Xena Toil project[101] at xenabrowser.net. The TCGA-LIHC study was used from that dataset. Survival data is available via the TCGA Pan-Cancer Clinical Data Resource (TCGA-CDR)[102] at the NIH Genomic Data commons (https://gdc.cancer.gov/about-data/publications/pancanatlas). Source data are provided with this paper.

## Code availability

Custom code used in this study has been described elsewhere[14]. Code used for processing TCGA data and producing Pearson's correlations is available at: https://github.com/Yenaled/felsher or https://doi.org/10.5281/zenodo.7643143[103].

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

## Acknowledgements

We thank members of the Felsher Laboratory for generously providing their suggestions and for thoughtful discussions. We thank Ian Lai for expert technical support. MRI was performed at the Stanford Center for Innovation in In-Vivo Imaging (SCi³) – small animal imaging center (supported by S10RR026917-01, PI Michael Moseley, PhD). Some of the results shown here are in whole or part based upon data generated by the TCGA Research Network: https://www.cancer.gov/tcga. This work was supported by the Lymphoma Research Foundation (A.D.), the Cancer Prevention and Research Institute of Texas (RP200472, Y.L.), the MIT Center for Precision Cancer Medicine (A.N.K.), and the following grants from the National Institute of Health: K22CA207598 (Y.L.), 1R35CA253180 (D.W.F.), R01CA089305 (D.W.F.), R01CA170378 (D.W.F) R01CA184384 (D.W.F.), U01CA188383 (D.W.F.), P50CA114747 (D.W.F.), and P30-CA14051 (A.N.K.).

## Author contributions

Y.L., A.D., and D.W.F. conceived and designed the study; A.D. and Y.L. performed CRISPR/Cas9 screening experiments; A.D., Y.L., D.K.S., W.L., and J.B. analyzed screen data; A.D., R.D., D.K.S., and W.M.F. performed and analyzed RNA-seq experiments. A.D. performed and analyzed the RNA imaging experiments. A.D., Y.L., and X.C. performed in vitro toxicity assays and Western blots. A.D. and R.D. performed PDX experiments. Y.L., A.D., L.T., and A.M. performed and analyzed the IHC and MRI experiments. D.K.S. and A.D. analyzed TCGA data. A.N.K. interpreted data and provided essential material. A.D., Y.L., and D.W.F. wrote, and A.D. and D.W.F. substantially revised the manuscript.

## Competing interests

A.N.K. is a founder of Kronos Bio and a member of its scientific advisory board. A.N.K. is a founder of 76Bio, serving on the scientific advisory board and board of directors. A.D., Y.L., and D.W.F. are inventors on a patent related to the method identifying MYC synthetic lethal interactions described in this work (Patent number 11576912). The remaining authors declare no competing interests.
