## [Peer Review File · Nature Communications]

Reviewers' comments:

Reviewer #1 (Remarks to the Author): Expertise in MYC (in vivo)

In this study by Li et al., the authors use a CRISPR-based genome-wide screening to determine essential genes in a murine cell line with switchable MYC expression. Thus, the authors identify a number of genes that are synthetically lethal only in the presence of high MYC, but not low MYC. Among them, they identified a gene set involved in nuclear to cytoplasmic transport as being significantly enriched. They further demonstrated that XPO1, a gene belonging to the aforementioned gene set, is an effective therapeutic target to treat Myc-driven HCC. Finally, they established the prognostic value of the various genes related to nuclear to cytoplasmic transport they identified and, specifically, of XPO1.

The manuscript is clear and well written. The experiments are also comprehensive and well-executed. Given that the main message of the manuscript seems to be the synthetic lethality of nuclear to cytoplasm transport genes and XPO1 as an example of these, there is one major concern:

The experiments described in this manuscript should demonstrate the advantage of inhibiting XPO1 in the context of high vs. low MYC protein levels. However, this comparison is between MYC-high and MYC-low contexts missing in some in vitro experiments (Supp Fig 6; Fig 2F), as well as in vivo experiments from Figures 3 and 4. Thus, while the use of XPO1 as a therapeutic target and prognostic marker in HCC are clearly demonstrated, this part of the paper lacks demonstration of efficacy and prognosis in a MYC-high vs. MYC-low context. This fact significantly impacts the novelty of the discovery, since XPO1 has been already targeted in various indications and has already been described recently as prognostic marker in PDAC (doi: 10.21037/sci.2019.02.03; doi: 10.3390/jcm8050596.). However, the reviewer acknowledges the novelty of describing nuclear to cytoplasm transport genes as potential synthetic lethal target in a MYC-high context. The reviewer leaves to the editor the decision of whether this discovery is sufficient for the ambition of the journal.

In opinion of this reviewer, the manuscript would be improved by the following revisions:

#1: Provide further proof of the synthetic lethality of XPO1 inhibition in a MYC-high vs. a MYC-low context

1.1: Supplementary Figure 6: ignoring the fact that human immortalized fibroblasts are used as control of mouse cancer cells, MYC protein levels of both cell lines should at least be compared by Western Blot. Ideally, the authors should use a MYC-low mouse cell line, just as they did for the CRISPR-based screening and in Figure 2E.

1.2: Same applies to Figure 2F. Also, a Western Blot/ qPCR showing knockdown efficiency is lacking. Only shRNA-positivity is not enough to prove that the observed phenotype is an outcome of targeted knockdown.

1.3: The authors should have performed in vivo experiments with MYC-high vs. MYC-low human cell lines xenograft models to prove that the targeting of XPO1 is only affecting MYC-high tumors. This is, however, a great amount of work and we leave this decision to the Editor.

#2: Line 23: 'is projected to increase by 35% by the year 2030'- increase in what? Incidence/prevalance? Deaths?

#3: Line 59: 'We defined as MYC-synthetic (MYC-SL) genes as genes...' Please, correct.

#4: line 80: is "nuclear to cytoplasmic transport" the only gene set identified that has not been previously related to MYC-SL?

#5: line 96: words missing? "... but not in MYC-low..."

#6: line 97: please reference Figure 1D.

#7: line 100: please provide examples of novel tumor suppressors identified.

#8: Line 104: the sentence 'the essentiality and MYC-regulation correlated' sounds very abrupt. Please explain in more detail.

#9: line 118: Inhibition of XPO1 clearly decreases ATP metabolism, indirectly reflecting cell viability, in EC4 cells with respect to BJ fibroblasts (Supp figure 6). This is, however, not indicative of induction of cell death as claimed by the authors. The authors must demonstrate the induction of cell death under these conditions to support their claim. Importantly, inhibition of PRMT5 doesn't have an effect on the proliferation of EC4 at concentrations that go beyond breast cancer cell lines IC50 (range: 0.8 – 3.9 μ M according to <https://www.selleckchem.com/products/epz015666.html>). Please, add fibroblasts to this panel. Also, as stated before, it would be important to show the response of MYC-low HCC cells.

#10: line 120: Provide justification for proceeding with the XPO1 inhibitor and not with the PRMT5 inhibitor. The authors can also comment on this in the discussion (line 188).

#11: line 122: Please correct: "Cell death induced by XPO1 inhibition of MYC-driven HCC depended upon high MYC expression levels..."

#12: line 124: "XPO1 inhibition blocked growth and induced death...". Cell death is not assessed in this experiment. Please correct.

#13: line 125: "...human P493-6 Burkitt lymphoma-like cell line...". This cell line is not shown in the corresponding SF8 figure.

#14: line 127: "...immortalized normal human fibroblast cells were not sensitive". At 1 μ M KPT-330 (fifth point), fibroblasts are close to 60% ATP levels. Even at the next lower concentration (fourth point, approximately 0.4 μ M), fibroblasts are close to 75%. Are these levels significantly different to control? If so, then they are less sensitive than EC4 cells.

#15: line 137: "... showed complete elimination of tumor" in 2 out of 6 mice. What time point was this analysis performed at? Additionally, the figures cited to support this observation (Figure 3d, supplementary Figure 10) do not demonstrate the author's claim.

#16: line 138: "... XPO1 inhibition did not induce changes..." Please correct. Also, provide quantification for the pH3 and cCASP3 IF in supplementary Figure 10.

#17: line 149: "...HCC patient-derived xenografts...". How many PDXs have been used (from more than 1 patient)? Are they MYC-high? Do the authors have evidence of it?

#18: line 624: "... cleaved-caspase 3 in liver tumor of mice after short-term treatment (3 doses)". Please, correct.

#19: Supplementary Fig 10 legend. "... Tumors and normal surrounding tissue sections were stained..." Please, correct.

#20: Other issues that can be included in the discussion:

9.1: outcome of XPO1 inhibitors in clinical trials.

9.2: The use of MYC levels as a predictive biomarker of response to XPO1 inhibition.

9.3: The discovery of XPO1 as a prognostic marker in other cancers.

Reviewer #2 (Remarks to the Author): Expertise in liver cancer and CRISPR screens

In this manuscript, the authors set up a genome-wide screen to discover genes that cancer cells depend upon in the setting of high levels of MYC, as compared to wild-type (low) levels of MYC, known as MYC-synthetic lethal (MYC-SL) genes. They cite four previous RNAi screens that were set up in a similar fashion. Cells used for the study were derived from their mouse model, a conditional MYC-off system wherein ectopic MYC is shut off when dox is added, as demonstrated nicely in a western blot in Fig S1. They show that essential genes tend to be depleted in both MYC-high and MYC-low conditions. They found 1,808 MYC-SL genes, using fairly loose criteria of significant depletion in MYC-on to a greater degree than MYC-off conditions. The genes that tend to be upregulated with MYC-ON also tend to be MYC-SL genes, indicating that a subset of MYC-regulated genes is required for cell survival. They perform pathway analysis and find RNA transport genes tend to be MYC-SL and are upregulated by MYC (Fig 2C). The findings of the screen and analysis are original and very interesting and the statistical analysis appears to be appropriate.

The authors then focus in on two genes that are involved in mRNA transport, are differentially regulated by MYC, and drugs exist that target them. In particular, they show functional data to target XPO1 using RNAi and drugs. For validation in vivo, the imaging data showing efficacy of XPO1i in mouse liver and PDX model is quite good, but showing gross images at the end of treatment or, ideally, survival data, would add rigor. XPO1 inhibitors are being used in clinical trials for different tumors, such as multiple myeloma and NSCLC. The clinical trial results are pretty encouraging. If the authors could prove that XPO1i is more effective in MYC-high tumors, it will be very helpful in providing a potential biomarker for patient stratification and targeted therapy. The authors should provide more information about whether XPO1i shows differences in MYC-high and MYC-low models.

Questions/suggestions:

1. What are the details of the gRNA abundance of XPO1 and PRMT5 before and after selection? Can the authors highlight XPO1 and PRMT5 in fig1B?
2. The authors mention 4 previous RNAi screens. Was XPO1 synthetic lethal in any of these previous screens?
3. How representative of human HCC is the EC4 cell line? Also, do the cell lines depend upon MYC expression to survive? If so, how does this influence the results of the screen? The discussion calls the MYC-low cells “hepatocyte-like”; how so?
4. More data about the effect of XPO1i in MYC high and low cells would be helpful. Such as with fig2f, the authors should show the data for both doxycycline+ and doxycycline- EC4 cells. In figure S6, the authors should also show the effect of PRMT5i and XPO1i in doxycycline+ EC4 cells. Also, why are BJ5-tA cells not shown for PRMT5i?
5. Figure4B, is it possible to examine whether patients with MYC (or MYC-signature) high and low subgroups have differential prognosis based on XPO1 expression level?
6. What is the expression level of MYC in the PDX model in figure 4?
7. It would add rigor to demonstrate the gross images of the tumors upon euthanasia in Fig 3, if available. Some additional data could be supportive of drug efficacy, such as the liver weight to body weight ratio, and perhaps quantification of the number of gross tumors in the mice.
8. S3 legend could use a more detailed description about why it is important to examine the distribution of read counts.
9. Minor typographical errors: line 138, missed “not” in “XPO1 inhibition did induce changes”; line596, should delete “of” in “Gene essentiality (Beta score) of is”.

Reviewer #3 (Remarks to the Author): Expertise in nuclear transport

The manuscript by Li et al. applies a CRISPR-based genetic screening approach to identify gene mutations that are synthetic lethal when the MYC protein is expressed. Generally, these data are interesting and provide more evidence for links between gene expression regulation, RNA transport, nuclear pore complexes, and cancer. However, the manuscript is largely descriptive in nature, and by focusing on XPO1, reports a link that is well established between XPO1 and many other cancers in both primary literature and more recent reviews. In this regard, a focus on other hits of the CRISPR screen may have been more interesting. Moreover, the text lacked detail, was abrupt in the presentation of results, and draws conclusions that appear overreaching and/or nebulous in origin based on the limited text. These problems prevented me from easily understanding the data, considering the conclusions being drawn, and appreciating novel aspects of the work. Given these issues, I would not support publication of the submitted manuscript.

Examples include:

1. The introduction, which was only one page in length, does not provide the background required to appreciate the results. For example, there is no discussion of XPO1 function, nor the well-established links between XPO1 and cancer. This info is also not discussed in any real detail or referenced well elsewhere in the text.
2. The use of MYC high vs. low is not accurate based on western blotting results in Fig S1 and I find this description to be confusing. Plus, in some instances of the text and figures it is referred to as MYC On and Off, which may be a more apt description of the system.
3. The results and text around the identification of essential genes vs. those that are SL with MYC-shutoff are presented together in the text via Fig 1c-e. This is very hard to follow. It is also unclear where the numbers come from out of figure 1d vs. 1e. For example, 1d would seem to suggest that there are 1221 genes that are essential when MYC is expressed based on the Venn diagram ($1858 - 587 = 1221$), but this is not what is discussed in the text.
4. It is unclear what the the sentence on line 104 is meant to convey? Similarly, Figure 2a-c is presented in 6 lines of text, which provides no framework for understanding the questions being addressed, the approaches, or why this was the question. This is generally an issue with much of the data being presented.
5. XPO1 is described as an RNA transport gene on page 111 onwards, which is at best a very incomplete statement. While XPO1 does support ncRNA export, and the export of select mRNAs, it is much more critical for protein transport. Yet, this aspect of XPO1 function is not discussed at all. The hit list does include Nxf1, Alyref, and many other proteins linked to mRNA processing and export, which may support their hypothesis, yet this is also not discussed.

RESPONSE TO REVIEWERS' COMMENTS

We thank the reviewers for their time and valuable comments to help us improve and strengthen our original manuscript significantly. We have updated the manuscript to address the reviewers' comments and questions.

A direct response to the reviewers' comments follows after a brief summary of new data and analyses we added to the manuscript. Our response is highlighted in blue and we use line numbers of the new draft and quotations to refer to changes we have made. In the revised manuscript, all edits are highlighted in blue.

Summary

Specifically, we have added the following **major new data and analyses** that directly address the reviewers' questions and suggestions:

1) Addition of a new HCC patient-derived xenograft (PDX) model. We found that the effect of XPO1 inhibition on relative tumor burden was more pronounced in a HCC PDX with high MYC expression compared to a PDX with low MYC expression (**new Figure 5d,e**).

2) Additional analysis of HCC TCGA data showing that XPO1 expression is prognostic in the group displaying high MYC activity while it has no prognostic power in the group with low MYC activity (new Figure 5b,c).

3) We added more detailed information on tumor suppressor genes identified in our CRISPR screen (new Figure 2).

4) We included additional in vitro toxicity data and show that MYC inhibition desensitizes cells toward XPO1 and PRMT5 inhibition (**new Supplementary Figure 7**).

5) We included new data to illustrate the effect of MYC downregulation in EC4 cells on proliferation, survival and gene expression. We provide new RNA expression data and gene set enrichment analyses in MYC^{high} and MYC^{low} states and show evidence for MYC inactivation-induced differentiation demonstrated by regulation of hepatocyte-specific genes (**new Supplementary Figure 3**).

Reviewer #1 (Remarks to the Author): Expertise in MYC (in vivo)

In this study by Li et al., the authors use a CRISPR-based genome-wide screening to determine essential genes in a murine cell line with switchable MYC expression. Thus, the authors identify a number of genes that are synthetically lethal only in the presence of high MYC, but not low MYC. Among them, they identified a gene set involved in nuclear to cytoplasmic transport as being significantly enriched. They further demonstrated that XPO1, a gene belonging to the aforementioned gene set, is an effective therapeutic target to treat Myc-driven HCC. Finally, they established the prognostic value of the various genes related to nuclear to cytoplasmic transport they identified and, specifically, of XPO1.

The manuscript is clear and well written. The experiments are also comprehensive and well-executed. Given that the main message of the manuscript seems to be the synthetic lethality of nuclear to cytoplasm transport genes and XPO1 as an example of these, there is one major concern: The experiments described in this manuscript should demonstrate the advantage of inhibiting XPO1 in the context of high vs. low MYC protein levels. However, this comparison is between MYC-high and MYC-low contexts missing in some in vitro experiments (Supp Fig 6; Fig 2F), as well as in vivo experiments from Figures 3 and 4. Thus, while the use of XPO1 as a therapeutic target and prognostic marker in HCC are clearly demonstrated, this part of the paper lacks demonstration of efficacy and prognosis in a MYC-high vs. MYC-low context. This fact significantly impacts the novelty of the discovery, since XPO1 has been already targeted in various indications and has already been described recently as prognostic marker in PDAC (doi: 10.21037/sci.2019.02.03; doi: 10.3390/jcm8050596.). However, the reviewer acknowledges the novelty of describing nuclear to cytoplasm transport genes as potential synthetic lethal target in a MYC-high context. The reviewer leaves to the editor the decision of whether this discovery is sufficient for the ambition of the journal. In opinion of this reviewer, the manuscript would be improved by the following revisions:

#1: Provide further proof of the synthetic lethality of XPO1 inhibition in a MYC-high vs. a MYC-low context.

We thank the reviewer for the suggestion to provide further proof synthetic lethality of XPO1 and MYC. We have now included further evidence that XPO1 activity and high MYC expression status/activity is a synthetic lethal interaction and that high MYC expression/activity sensitizes to XPO1 inhibition. First, we have included additional PDX data showing efficacy of XPO1 inhibition on HCC PDX growth in the context of high and low MYC expression (**Figure 5d,e**).

Second, we provide toxicity data showing differential efficacy to XPO1 inhibitor KPT-330 in the context of MYC inhibition via MAX/MAX stabilization by MS2-008 (**Supplementary Figure 7c,d**, and see also comment 1.1 below).

1.1: Supplementary Figure 6: ignoring the fact that human immortalized fibroblasts are used as control of mouse cancer cells, MYC protein levels of both cell lines should at least be compared by Western Blot. Ideally, the authors should use a MYC-low mouse cell line, just as they did for the CRISPR-based screening and in Figure 2E.

We agree with the reviewer that murine instead of human fibroblasts are the better control for *in vitro* efficacy studies of XPO1 inhibition in murine HCC cells. We have added human immortalized fibroblast and murine fibroblast toxicity data for XPO1 and PRMT5 inhibitors and have also provided Western blot analysis of MYC expression in the cell lines used (**Supplementary Figure 7a,b**).

To compare states of high and low MYC activity as suggested by the reviewer, we have tested the sensitivity of human HCC cells with high MYC level (SNU-449) to XPO1 or PRMT5 inhibition in the presence or absence of MYC inhibition using the MAX/MAX stabilizer MS2-008. We show that MYC inhibition confers decreased sensitivity to both XPO1 and PRMT5 inhibition in human HCC cells (**Supplementary Figure 7c,d**).

1.2: Same applies to Figure 2F. Also, a Western Blot/ qPCR showing knockdown efficiency is lacking. Only shRNA-positivity is not enough to prove that the observed phenotype is an outcome of targeted knockdown.

We have provided a Western Blot showing knockdown efficiency in the supplementary information (**Supplementary Figure 8b**).

1.3: The authors should have performed *in vivo* experiments with MYC-high vs. MYC-low human cell lines xenograft models to prove that the targeting of XPO1 is only affecting MYC-high tumors. This is, however, a great amount of work and we leave this decision to the Editor.

We agree with the reviewer that additional *in vivo* evidence for XPO1 inhibition efficacy in the context of varying MYC expression would strengthen our finding. To this end, we have established a new HCC patient-derived xenograft and show that increased MYC expression is concomitant with increased sensitivity to Xpo1 inhibition (**Figure 5d,e**). However, we would like to point out that in our primary mouse model of HCC the efficacy of XPO1 inhibitor KPT-330 was more pronounced compared to xenograft models. We speculate that XPO1 inhibition-induced tumor cell death leads to changes in the tumor microenvironment and possibly activation of an immune response which is lacking in xenograft models. We believe that xenograft models can lead to underestimation of efficacy and differential response.

#2: Line 23: 'is projected to increase by 35% by the year 2030'- increase in what? Incidence/prevalance? Deaths?

We have corrected this sentence to: "The number of new HCC cases each year is projected to increase by 35% by the year 2030⁸."

#3: Line 59: 'We defined as MYC-synthetic (MYC-SL) genes as genes...' Please, correct.

We have changed this sentence to (**lines 61-67**): "Genes that had a significant (false discovery rate-adjusted p value (FDR) < 0.05) negative Beta score were considered essential. We defined genes as a MYC synthetic lethal (MYC-SL) interaction, those causing cell death or significant proliferation deficits only in cells with high MYC levels, if the knockout resulted in 1) a negative Beta score indicative of negative selection of gRNAs targeting these genes only in the MYC^{high} condition (FDR < 0.05) and 2) no significant change in cell fitness of the MYC^{low} control cells (FDR > 0.05, or Beta score > 0) (**Figure 1b**)."

#4: line 80: is "nuclear to cytoplasmic transport" the only gene set identified that has not been previously related to MYC-SL?

To our knowledge, we are reporting the first comprehensive genome-wide analysis of MYC synthetic lethal interactions including detailed analysis of pathway enrichment. However, specific MYC synthetic lethal genes or single pathways, such as ribosomal biogenesis, have been described in the past by multiple groups (see lines 83-85). While the RNA transport pathway is not the only gene set that is statistically enriched in the MYC synthetic lethal genes we identified, we found that expression of genes belonging to this particular pathway is positively regulated in a MYC-dependent manner. We speculate

that MYC-induced upregulation of RNA transport amplifies the synthetic lethal interaction and contributes to therapeutic efficacy of RNA transport inhibition in cancers with high MYC levels.

#5: line 96: words missing? "... but not in MYC^{low}..."

We have corrected this sentence to (lines 105-107): "We identified 1,543 genes with antiproliferative function only in MYC^{high} but not in MYC^{low} cells, and 173 genes with antiproliferative function in both MYC^{high} and MYC^{low} cells (Figure 2a)."

#6: line 97: please reference Figure 1D.

We have now referenced Figure 1d in line 84.

#7: line 100: please provide examples of novel tumor suppressors identified.

We have now provided more information on the tumor suppressor candidates identified (see new Figure 2).

We have changed lines 108-112 to include identified Hippo signaling pathway components or regulators (Lats1, Nf2, Sav2, Amotl2, Snai2, Pals1, Vgll4). We have also added apolipoprotein B (APOB) as an example of a novel tumor suppressor, which we found to have antiproliferative function in both MYC^{high} and MYC^{low} conditions. APOB is mutated in about 10% of human HCC and APOB mutation is associated with decreased survival probability in HCC (new Figure 2e, lines 112-118).

We have also added the following to the discussion (lines 235-241): "Our screen also uncovered tumor suppressor genes whose loss cooperates with MYC to promote cancer. Examples for identified well-known tumor suppressors include genes such as *Trp53*, *Rb1*, *Nf1* and *Nf2*, and Hippo signaling pathway components. However, other genes with similar tumor suppressive function in our screen, such as *Ambra1*, *Gpatch8*, and *Spred1*, may be less appreciated as tumor suppressors. Thus, our genome wide CRISPR-screen served as a powerful approach to determine specific MYC-SL genes and pathways."

#8: Line 104: the sentence 'the essentiality and MYC-regulation correlated' sounds very abrupt. Please explain in more detail.

We have now explained our reasoning in more detail and have replaced the sentence by (**lines 120-127**): "We hypothesized that MYC-SL genes, which are most strongly induced by MYC, would be the best therapeutic targets for MYC-driven HCC. Therefore, we performed differential gene expression analysis on primary MYC^{high} and MYC^{low} HCC tumors to identify MYC-driven gene expression changes *in situ*. We then assessed whether the Beta score of the 1,808 identified MYC-SL genes and their MYC-driven gene expression changes correlated. We found that gene expression stimulated by MYC was associated with MYC-SL gene essentiality in the MYC^{high} condition (**Figure 3a**)."

#9: line 118: Inhibition of XPO1 clearly decreases ATP metabolism, indirectly reflecting cell viability, in EC4 cells with respect to BJ fibroblasts (Supp figure 6). This is, however, not indicative of induction of cell death as claimed by the authors. The authors must demonstrate the induction of cell death under these conditions to support their claim. Importantly, inhibition of PRMT5 doesn't have an effect on the proliferation of EC4 at concentrations that go beyond breast cancer cell lines IC50 (range: 0.8 – 3.9µM according to <https://www.selleckchem.com/products/epz015666.html>). Please, add fibroblasts to this panel. Also, as stated before, it would be important to show the response of MYC-low HCC cells.

We thank the reviewer for this attentive comment. We agree with the reviewer that ATP levels are no direct indication of viability as cells can be less metabolically active while still alive. That is why we have demonstrated that XPO1 inhibition induced cell death by flow cytometric analysis of EC4 cells with high or low MYC expression using PI/AnnexinV staining (**Figure 3e, see also Supplementary Figure 8a**). We have repeated the ATP assay for XPO1 and PRMT5 inhibitors in HCC cell lines (human SNU449, murine EC4) and fibroblasts (human BJ5-tA, murine NIH3T) and have included this data in **Supplementary Figure 7b**. EC4 cells are sensitive to both XPO1 and PRMT5 inhibition. Importantly, we have now also compared sensitivity of human HCC cells with high MYC level to XPO1 or PRMT5 inhibition in the context of MYC inhibition. In addition to ATP levels as proxy for cell viability, we have now included sulforhodamine B assay data reflecting cell density/ biomass via protein content measurement (**Supplementary Figure 7c,d**). Both, XPO1 and PRMT5 inhibition were more effective in the context of high MYC activity confirming their MYC synthetic lethal function. We have added the following sentence to the main manuscript (**lines 145-148**): "MYC-driven HCC cells exhibited decreased ATP levels in response to the inhibition of XPO1 and PRMT5 and fibroblasts were less sensitive towards XPO1 and PRMT5 inhibition than MYC-driven HCC cells (**Supplementary Figure 7a,b**)."

#10: line 120: Provide justification for proceeding with the XPO1 inhibitor and not with the PRMT5 inhibitor. The authors can also comment on this in the discussion (line 188).

The dependency of MYC-driven cancer on proper spliceosome function had been described (Hsu et al., Nature, 2015), and PRMT5 as well as SMN complex components identified as MYC synthetic-lethal in our screen perform essential roles in RNA splicing. While PRMT5 activity affects RNA metabolism and transport, PRMT5 might also exert its tumorigenic properties via activation of other oncogenic pathways by methylation of certain target proteins, such as p53 (Jansson et al., Nature Cell Biology, 2015). We recently showed that MYC via upregulation of and cooperation with SREBP1 drives lipogenesis essential for tumor growth (Gouw et al., Cell Metabolism, 2019). Interestingly, PRMT5 was shown to methylate SREBP1 leading to increased protein stability and increased lipogenesis (Liu et al., Cancer Research, 2016). However, nucleocytoplasmic transport had not been appreciated as a

therapeutic vulnerability in MYC-driven cancer. We chose to investigate XPO1 as an example for a MYC synthetic-lethal interaction with direct nucleocytoplasmic transport function, now stated in lines 154-155. The XPO1 inhibitor KPT-330 (Selinexor) showed promising efficacy in multiple preclinical tumor models and already advanced to clinical testing at the time of our initial screen results and was the only small molecule inhibitor directly targeting nucleocytoplasmic transport.

#11: line 122: Please correct: “Cell death induced by XPO1 inhibition of MYC-driven HCC depended upon high MYC expression levels...”

We have corrected this sentence to now say (lines 157-159): “XPO1 inhibition increased cell death by 4.4-fold in MYC-driven HCC with high MYC expression levels while hardly affecting survival of cells with low MYC levels (Figure 3e, Supplementary Figure 8a).”

#12: line 124: “XPO1 inhibition blocked growth and induced death...”. Cell death is not assessed in this experiment. Please correct.

We have corrected this sentence to (lines 159-162): “XPO1 inhibition decreased cell fitness in other MYC-driven tumors including human P493-6 Burkitt lymphoma-like cell line, a murine MYC-induced T-cell leukemia cell line and a murine *IgH-MYC* B-cell lymphoma cell line (Supplementary Figure 9).”

#13: line 125: “...human P493-6 Burkitt lymphoma-like cell line...”. This cell line is not shown in the corresponding SF8 figure.

We have corrected Supplementary Figure 9 to include P493-6 cells instead of BJ5tA fibroblasts. The effect of XPO1 inhibition on fibroblasts can now be found in Supplementary Figure 7b.

#14: line 127: “...immortalized normal human fibroblast cells were not sensitive”. At 1 μM KPT-330 (fifth point), fibroblasts are close to 60% ATP levels. Even at the next lower concentration (fourth point, approximately 0.4 μM), fibroblasts are close to 75%. Are these levels significantly different to control? If so, then they are less sensitive than EC4 cells.

As pointed out by the reviewer, human fibroblasts were sensitive to high concentrations of KPT-330. EC₅₀ concentrations were significantly different in EC4 cells compared to murine fibroblasts for both inhibitors. Human HCC cell line SNU-449 was significantly more sensitive to XPO1 inhibition than compared to human fibroblasts. However, the sensitivity of SNU-449 cells and human fibroblasts to PRMT5 inhibition was not significantly different. We have now summarized this data in Supplementary Figure 7 and have added the following statement to the manuscript (lines 145-153): “MYC-driven HCC cells exhibited decreased ATP levels in response to the inhibition of XPO1 and PRMT5 and fibroblasts were less sensitive towards XPO1 and PRMT5 inhibition than MYC-driven HCC cells (Supplementary Figure 7a,b). In the human HCC cell line SNU-449, blocking MYC activity using the MAX/MAX homodimer stabilizer MS2-008⁵⁷ mitigated the effects of both XPO1 and PRMT5 inhibition on ATP levels (Supplementary Figure 7c) and led to a modest increase in bio mass accumulation at high inhibitor concentrations compared to control (Supplementary Figure 7d). Hence, there appears to be a synthetic lethal interaction between MYC and XPO1, and MYC and PRMT5.”

#15: line 137: "... showed complete elimination of tumor" in 2 out of 6 mice. What time point was this analysis performed at? Additionally, the figures cited to support this observation (Figure 3d, supplementary Figure 10) do not demonstrate the author's claim.

Tumors were undetectable by MRI in two out of six Selinexor treated mice and we have now cited the correct figure (**Figure 4b**). More representative MRI images are shown in Supplementary Figure 10. Post-treatment MRI was performed after two weeks of XPO1 inhibitor treatment (3 doses per week for two weeks). We have added this information to the figure legend and rephrased the results to now say: "We assessed liver tumor volume by magnetic resonance imaging (MRI) before and after treatment (**Supplementary Figure 10**). XPO1 inhibition (3 doses per week for two weeks) resulted in a greater than 95% decrease in tumor volume compared with a 4-12-fold increase in tumor size in vehicle control mice (**Figure 4a-c**). In two out of six Selinexor-treated mice, tumors were undetectable by MRI post treatment." Figure 4d-f show histological data upon short-term treatment (3 doses) and tissue was harvested and processed on day 7 post treatment start. We have replaced "...showed complete elimination of tumor" by: "Histological examination after short-term treatment showed that residual abnormal tissue did not resemble tumor tissue but rather necrotic tissue with decreased nuclear to cytoplasmic ratio."

#16: line 138: "... XPO1 inhibition did not induce changes..." Please correct. Also, provide quantification for the pH3 and cCASP3 IF in supplementary Figure 10.

We have corrected the sentence to: "In contrast, XPO1 inhibition did not induce changes in proliferation or apoptosis in adjacent normal liver tissue (**Supplementary Figure 11**)."

We have also now included a quantification of pH3 and cCasp3 immunofluorescence stainings in **Supplementary Fig 11c**.

#17: line 149: "...HCC patient-derived xenografts...". How many PDXs have been used (from more than 1 patient)? Are they MYC-high? Do the authors have evidence of it?

In the initial manuscript, the PDXs were generated from one patient sample. MYC was confirmed on protein level in the primary tumor (PDX-37). We have now added data (see **Figure 5d,e**) on a second PDX cohort originating from a different patient sample that had significantly higher MYC levels (PDX-58). XPO1 inhibition slowed down tumor progression in both xenografts and higher MYC levels appeared to confer increased sensitivity towards XPO1 inhibition.

#18: line 624: "... cleaved-caspase 3 in liver tumor of mice after short-term treatment (3 doses)". Please, correct.

We have corrected this sentence to (**lines 766-768**): "Immunofluorescence staining and quantification of phospho-histone H3 and cleaved-caspase 3 in liver tumors of mice after short-term treatment with vehicle or Selinexor (3 doses per week, one week)."

#19: Supplementary Fig 10 legend. "... Tumors and normal surrounding tissue sections were stained..." Please, correct.

We have corrected the figure legend (**Supplementary Figure 11b**) to "b) Representative immunofluorescence images of tumor-adjacent normal liver tissue stained for phospho-histone H3 (pH3) and cleaved-caspase 3 (cCasp3) of mice described in a). Cell nuclei were counterstained with DAPI."

#20: Other issues that can be included in the discussion:

We have now included more information and thoughts on XPO1 as a therapeutic target in MYC-driven cancers in the discussion as outlined below.

9.1: outcome of XPO1 inhibitors in clinical trials.

Lines 282-285: "Selinexor is a highly selective XPO1 inhibitor which has been FDA approved for the treatment of refractory multiple myeloma and diffuse large B-cell lymphoma, and is being currently evaluated in clinical trials for other cancers (NCT02606461, NCT03555422, NCT02649790).

9.2: The use of MYC levels as a predictive biomarker of response to XPO1 inhibition.

And 9.3: The discovery of XPO1 as a prognostic marker in other cancers.

MYC activity as predictive biomarker of response to XPO1 inhibition:

Lines 278-282: "Based on the synthetic lethal interaction of MYC and XPO1 described here, we demonstrate that XPO1 inhibition was most effective in tumors with high MYC activity. In line with this, in human HCC, XPO1 levels were predictive of clinical outcomes only in tumors with high, but not low, expression of MYC activation signature."

Lines 286-290: "Our study demonstrates that stratification of tumors based on MYC activation status will likely enable more precise selection of patients for RNA transport inhibitors. More importantly, we identify several MYC-SL genes in the RNA transport pathway, apart from XPO1, which are yet to be targeted and can serve as rational, and specific targets against MYC-driven cancers."

XPO1 as prognostic marker:

Lines 252-254: "High XPO1 expression has been associated with poor prognosis in multiple cancers including ovarian cancer⁷⁰, osteosarcoma⁷¹, acute myeloid leukemia⁷², pancreatic adenocarcinoma⁷³, and neuroblastoma⁷⁴."

Reviewer #2 (Remarks to the Author): Expertise in liver cancer and CRISPR screens

In this manuscript, the authors set up a genome-wide screen to discover genes that cancer cells depend upon in the setting of high levels of MYC, as compared to wild-type (low) levels of MYC, known as MYC-synthetic lethal (MYC-SL) genes. They cite four previous RNAi screens that were set up in a similar fashion. Cells used for the study were derived from their mouse model, a conditional MYC-off system wherein ectopic MYC is shut off when dox is added, as demonstrated nicely in a western blot in Fig S1. They show that essential genes tend to be depleted in both MYC-high and MYC-low conditions. They found 1,808 MYC-SL genes, using fairly loose criteria of significant depletion in MYC-on to a greater degree than MYC-off conditions. The genes that tend to be upregulated with MYC-ON also tend to be MYC-SL genes, indicating that a subset of MYC-regulated genes is required for cell survival. They perform pathway analysis and find RNA transport genes tend to be MYC-SL and are upregulated by MYC (Fig 2C). The findings of the screen and analysis are original and very interesting and the statistical analysis appears to be appropriate.

The authors then focus in on two genes that are involved in mRNA transport, are differentially regulated by MYC, and drugs exist that target them. In particular, they show functional data to target XPO1 using RNAi and drugs. For validation in vivo, the imaging data showing efficacy of XPO1i in mouse liver and PDX model is quite good, but showing gross images at the end of treatment or, ideally, survival data, would add rigor. XPO1 inhibitors are being used in clinical trials for different tumors, such as multiple myeloma and NSCLC. The clinical trial results are pretty encouraging. If the authors could prove that XPO1i is more effective in MYC-high tumors, it will be very helpful in providing a potential biomarker for patient stratification and targeted therapy. The authors should provide more information about whether XPO1i shows differences in MYC-high and MYC-low models.

Questions/suggestions:

1. What are the details of the gRNA abundance of XPO1 and PRMT5 before and after selection? Can the authors highlight XPO1 and PRMT5 in fig1B?

gRNA abundance was only assessed after selection as the main question was the effect of MYC expression status on baseline gRNA abundance. gRNA abundances for XPO1 and PRMT5 in the selected baseline pool was comparable to the overall frequency distribution of all gRNAs.

XPO1 and PRMT5 beta scores (in MYC^{high} vs. MYC^{low}) fall within a cluster of MYC-SL genes in the genome-wide representation of beta scores in Figure 1b (highlighted in red). To better visualize beta scores of XPO1 and PRMT5 as suggested by the reviewer, we have now added a dot plot (**Supplementary Figure 6**) that only contains beta scores of MYC-SL RNA transport genes that are significantly upregulated by MYC as shown in Figure 3b,c.

2. The authors mention 4 previous RNAi screens. Was XPO1 synthetic lethal in any of these previous screens?

To our knowledge, XPO1 has not been identified as a MYC synthetic-lethal gene in previous RNA interference screens and nucleocytoplasmic transport has not been described as a MYC synthetic lethal pathway.

3. How representative of human HCC is the EC4 cell line? Also, do the cell lines depend upon MYC expression to survive? If so, how does this influence the results of the screen? The discussion calls the MYC-low cells “hepatocyte-like”; how so?

The murine HCC cell line EC4 was generated from liver tumors of LAP-tTA/TRE-MYC mice which we have reported previously to develop tumors with human HCC features (Beer et al., Shachaf et al.).

We have provided new data to demonstrate that, within the time frame of our screening experiment (7 days of MYC downregulation), downregulation of MYC expression in EC4 cells does not significantly affect cell viability but slows down proliferation (Supplementary Figure 3a, see below).

We have added new RNA expression data and gene set enrichment analyses to demonstrate the induction of a hepatocyte-like phenotype upon MYC downregulation in EC4 cells which leads to upregulation of hepatocyte-specific genes (Supplementary Figure 3b-e).

c

Gene set	NES	p-value	FDR
AIZARANI_LIVER_C11_HEPATOCYTES_1	1.44	0.00	0.02
AIZARANI_LIVER_C14_HEPATOCYTES_2	1.48	0.00	0.02
AIZARANI_LIVER_C17_HEPATOCYTES_3	1.43	0.01	0.01
AIZARANI_LIVER_C30_HEPATOCYTES_4	1.25	0.02	0.07

We have added the following statement in the results section (**lines 43-49**): “MYC downregulation in EC4-Cas9 cells induced an initial reduction in proliferation but did not affect viability (**Supplementary Figure 3a**). Upon MYC downregulation for one week, EC4 cells become hepatocyte-like based on the observed enrichment of expression of hepatocyte-specific genes (**Supplementary Figure 3b-e**), that we considered as the MYC^{low} control. This result is in line with our previous findings that MYC induces dedifferentiation¹⁴ and that MYC downregulation can induce a differentiation phenotype in HCC¹³.”

4. More data about the effect of XPO1i in MYC high and low cells would be helpful. Such as with fig2f, the authors should show the data for both doxycycline+ and doxycycline- EC4 cells. In figure S6, the authors should also show the effect of PRMT5i and XPO1i in doxycycline+ EC4 cells. Also, why are BJ5-tA cells not shown for PRMT5i?

We have now included efficacy data (ATP levels) for XPO1 and PRMT5 inhibition in murine (EC4, NIH3T3) and human (SNU-449, BJ5-tA) HCC cells and fibroblasts (**Supplementary Figure 7a,b**).

To compare states of high and low MYC activity as suggested by the reviewer, we have tested the sensitivity of human HCC cells with high MYC level (SNU-449) to XPO1 or PRMT5 inhibition in the presence or absence of MYC inhibition using the MAX/MAX stabilizer MS2-008. We show that MYC inhibition confers decreased sensitivity to both XPO1 and PRMT5 inhibition in human HCC cells (**Supplementary Figure 7c,d**).

5. Figure 4B, is it possible to examine whether patients with MYC (or MYC-signature) high and low subgroups have differential prognosis based on XPO1 expression level?

We thank the reviewer for this thoughtful suggestion. While MYC expression alone is not prognostic in the TCGA-LIHC data set (data not shown), survival probability can be stratified by expression of MYC hallmark genes. We have included a new figure that shows XPO1 expression level is prognostic in HCC patients with high MYC signature expression and does not have prognostic value in patients with low MYC signature expression (**Figure 5b,c**).

We have added the following text to the corresponding results section (**lines 191-201**): “To examine whether HCC patients with high and low MYC activity have different prognoses depending on XPO1 expression, we stratified the TCGA-LIHC cohort by expression of MYC-regulated hallmark genes and evaluated the effect of XPO1 expression on survival probability in these groups. Overexpression of MYC hallmark genes in human HCC patients predicted decreased overall survival (**Figure 5b**). In patients with high MYC activity (MYC_Hallmarks_V2_score^{high}), XPO1 had prognostic power and high XPO1 expression predicted poor outcome. However, XPO1 expression had no prognostic power in patients with low MYC activity (MYC_Hallmarks_V2_score^{low}) (**Figure 5c**). Therefore, MYC-SL RNA transport genes and XPO1 specifically predict poor prognosis, and XPO1 is a prognostic biomarker only in cancers with high MYC activity.”

6. What is the expression level of MYC in the PDX model in figure 4?

For the initial submission, the PDXs were generated from one patient sample. MYC was confirmed on protein level in the primary tumor (PDX-37), however, MYC expression was low. We have now added data (see **Figure 5d,e**) on a second PDX cohort originating from a different patient sample that had significantly higher MYC levels (PDX-58). XPO1 inhibition slowed down tumor progression in both xenograft models. PDXs with higher MYC levels appeared to be more sensitive to XPO1 inhibition.

7. It would add rigor to demonstrate the gross images of the tumors upon euthanasia in Fig 3, if available. Some additional data could be supportive of drug efficacy, such as the liver weight to body weight ratio, and perhaps quantification of the number of gross tumors in the mice.

While additional descriptive measures would support the data shown in Figure 4, we believe that MRI imaging data is quantitative and allows to determine before and after treatment tumor volumes accurately. Figures of gross tumors and liver and body weight data are not available for this cohort.

8. S3 legend could use a more detailed description about why it is important to examine the distribution of read counts.

We have expanded the figure legend of **Supplementary Figure 4** to: “Distribution of read counts was assessed as a quality control metric to confirm consistent gRNA abundance distribution between samples. The frequencies of all gRNA read counts were determined and visualized by MAGeCK-VISPR in three conditions (MYC^{high}, MYC^{low}, and baseline). Similar distributions across samples allowed for sample comparison.”

9. Minor typographical errors: line 138, missed “not” in “XPO1 inhibition did induce changes”; line 596, should delete “of” in “Gene essentiality (Beta score) of is”.

We have corrected these sentences accordingly to:

Lines 178-180: “In contrast, XPO1 inhibition did not induce changes in proliferation or apoptosis in adjacent normal liver tissue (**Supplementary Figure 11**).”

Lines 736-738: “Gene essentiality (Beta score) is plotted for genes that are either significantly up or down regulated in MYC^{high} compared to MYC^{low} HCC (adjusted p value < 0.05).”

Dingzi Yin, Kirk J. Wangenstein
Departments of Medicine and Genetics
University of Pennsylvania

Reviewer #3 (Remarks to the Author): Expertise in nuclear transport

The manuscript by Li et al. applies a CRISPR-based genetic screening approach to identify gene mutations that are synthetic lethal when the MYC protein is expressed. Generally, these data are interesting and provide more evidence for links between gene expression regulation, RNA transport, nuclear pore complexes, and cancer. However, the manuscript is largely descriptive in nature, and by focusing on XPO1, reports a link that is well established between XPO1 and many other cancers in both primary literature and more recent reviews. In this regard, a focus on other hits of the CRISPR screen may have been more interesting. Moreover, the text lacked detail, was abrupt in the presentation of results, and draws conclusions that appear overreaching and/or nebulous in origin based on the limited text. These problems prevented me from easily understanding the data, considering the conclusions being drawn, and appreciating novel aspects of the work. Given these issues, I would not support publication of the submitted manuscript.

Examples include:

1. The introduction, which was only one page in length, does not provide the background required to appreciate the results. For example, there is no discussion of XPO1 function, nor the well-established links between XPO1 and cancer. This info is also not discussed in any real detail or referenced well elsewhere in the text.

We thank the reviewer for the suggestion to provide additional information on XPO1 function and its role in cancer to allow the reader to appreciate our study results. This study is based on a genome-wide screening approach leading up to the identification of the RNA transport pathway as a MYC-SL interaction, with XPO1 as an example for an RNA transport gene with essential function in MYC-driven HCC. Thus, we believe that the introduction should rather focus on our unbiased comprehensive screening approach we used to identify MYC synthetic-lethal interactions. To address the reviewer's suggestion, we have significantly expanded our discussion to provide background on XPO1 function and role in cancer and information on other RNA transport genes identified as MYC synthetic lethal interactions.

2. The use of MYC high vs. low is not accurate based on western blotting results in Fig S1 and I find this description to be confusing. Plus, in some instances of the text and figures it is referred to as MYC On and Off, which may be a more apt description of the system.

We thank the reviewer for pointing out the discrepancy in figure labeling and figure legends. The reviewer correctly points out that our system allows for virtual shut off of expression of the human MYC transgene. However, the screening cell line expresses endogenous murine MYC at low levels and to reflect this, we prefer to refer to cells with downregulated transgene expression as "MYC^{low}" cells (rather than "MYC^{off}"). Interestingly, knockout of endogenous murine MYC is lethal for cells in the MYC^{low} condition while it has not effect on MYC^{high} cells, illustrating the importance of low MYC levels for survival of control cells. We have corrected MYC On/Off labels (Figure 1a, Figure 3, and corresponding legends) to now read MYC^{high} and MYC^{low}.

3. The results and text around the identification of essential genes vs. those that are SL with MYC-shutoff are presented together in the text via Fig 1c-e. This is very hard to follow. It is also unclear where the numbers come from out of figure 1d vs. 1e. For example, 1d would seem to suggest that there are 1221 genes that are essential when MYC is expressed based on the Venn diagram (1858-587 = 1221), but this is not what is discussed in the text.

d

We have moved the labels of the Venn diagram inside the diagram and added boundaries to improve visualization (**Figure 1d**). To clarify further, we have edited the corresponding results section to state (**lines 61-79**): “Genes that had a significant (false discovery rate-adjusted p value (FDR) < 0.05) negative Beta score were considered essential. We defined genes as a MYC synthetic lethal (MYC-SL) interaction, those causing cell death or significant proliferation deficits only in cells with high MYC levels, if the knockout resulted in 1) a negative Beta score indicative of negative selection of gRNAs targeting these genes only in the MYC^{high} condition (FDR < 0.05) and 2) no significant

change in cell fitness of the MYC^{low} control cells (FDR > 0.05, or Beta score > 0) (**Figure 1b**). We identified 2395 genes that are required for the survival of HCC MYC^{high} tumor cells, 682 genes with essential function in MYC^{low} cells. Genes that were essential in both MYC^{low} and MYC^{high} conditions (587 genes) were considered to be required for proliferation and survival irrespective of the level of MYC expression. Gene set enrichment analysis (GSEA) showed that genes, which were previously identified to be essential in human cancer cell lines¹⁷, were preferentially depleted in both MYC^{high} and MYC^{low} conditions while non-essential genes¹⁷ were not (**Figure 1c**), illustrating the validity of our screen. Moreover, genes that have been described to be essential in mouse and human¹⁸ were preferentially depleted in both MYC^{high} and MYC^{low} conditions (**Figure 1c**) which is in agreement with previous studies. Therefore, 1,808 MYC-SL genes were identified that had essential functions in MYC^{high} but not MYC^{low} tumor cells. (**Figure 1d, Supplementary Table 1**).”

4. It is unclear what the the sentence on line 104 is meant to convey? Similarly, Figure 2a-c is presented in 6 lines of text, which provides no framework for understanding the questions being addressed, the approaches, or why this was the question. This is generally an issue with much of the data being presented.

We thank the reviewer for pointing out how a more detailed framework would allow a better understanding of our rationale for performing correlation analysis between MYC-SL gene essentiality and MYC-driven gene regulation. We have expanded the results paragraph to provide more context for the question addressed in (now) Figure 3 (**lines 120-127**): “We hypothesized that MYC-SL genes, which are most strongly induced by MYC, would be the best therapeutic targets for MYC-driven HCC. Therefore, we performed differential gene expression analysis on primary MYC^{high} and MYC^{low} HCC tumors to identify MYC-driven gene expression changes *in situ*. We then assessed whether the Beta scores of the 1,808 identified MYC-SL genes and their MYC-driven gene expression changes correlated. We found that gene expression stimulated by MYC was associated with MYC-SL gene essentiality in the MYC^{high} condition (**Figure 3a**).”

In addition, we have also expanded other result sections to include more context for the hypotheses and questions that are being addressed.

5. XPO1 is described as an RNA transport gene on page 111 onwards, which is at best a very incomplete statement. While XPO1 does support ncRNA export, and the export of select mRNAs, it is much more critical for protein transport. Yet, this aspect of XPO1 function is not discussed at all. The

hit list does include Nxf1, Alyref, and many other proteins linked to mRNA processing and export, which may support their hypothesis, yet this is also not discussed.

We thank the reviewer for the suggestion to discuss XPO1 function and other identified MYC synthetic-lethal interactions with roles in RNA processing and export more comprehensively. As the reviewer mentions, we have identified additional MYC-SL genes involved in RNA transport and RNA metabolism that are MYC synthetic lethal which supports our finding that XPO1-mediated nuclear to cytoplasmic transport is a vulnerability of MYC-driven HCC. To better illustrate the importance of our findings, we have now added additional information to the results section and have expanded the discussion of XPO1 function and included context for other identified MYC synthetic-lethal genes that play important roles in RNA processing and nucleocytoplasmic transport.

Lines 126-136: “We found that gene expression stimulated by MYC was associated with MYC-SL gene essentiality in the MYC^{high} condition (**Figure 3a**). Pathway analysis of the 516 MYC-SL genes that are upregulated by MYC identified 47 RNA transport genes that are involved in RNA metabolism, mRNA surveillance and splicing, and nuclear to cytoplasmic transport of RNA (**Figure 3b,c, Supplementary Figure 6, Supplementary Tables 5-7**). These genes are involved in splicing-coupled mRNA/mRNP export like the TREX complex components (ALYREF, THOC1, THOC3), general mRNA export receptors (NXF1, exportins XPO1 and XPO5), and components of the nuclear pore complex itself. This suggests that MYC regulates the expression of specific RNA transport genes required for nuclear to cytoplasmic transport and also that these genes are essential in MYC-driven tumors.”

Lines 250-263: “One of the nuclear exporters identified as MYC-SL in our screen was XPO1, a gene involved in the transport of thousands of proteins, ribosomal RNAs and less abundant RNA species⁶⁹. [...] Apart from XPO1, our screen also identified other proteins transported by XPO1 to be products of MYC-SL genes. Interestingly, several of these proteins are also known to be important in MYC-driven cancers like ribosome components (such as RPS8), components of the eukaryotic translation initiation factor 3 (eIF-3) complex, and proteins important for MYC-induced autophagy (such as ATG3). Taken together, these findings highlight the critical role of XPO1 mediated nuclear to cytoplasmic transport in promoting MYC-driven tumor growth.”

Lines 265-275: “We show here that several genes involved in mRNA processing and transport play essential roles in MYC-driven cancer cells. These included components of the exon junction complex (such as *ALYREF*, *PININ*, *RNPS1*, and *DDX39B*), transcription export complex (such as *ALYREF*, *THOC1*, and *THOC3*), and the major mRNA exporter *NXF1*. An interesting MYC-SL gene we identified was *NMD3*, a 60S ribosomal export protein which together with *XPO1*, transports ribosomal subunits from the nucleus to the cytoplasm⁷⁵⁻⁷⁷. The finding that multiple genes involved in RNA transport, by a XPO1-independent or -dependent mechanism, have synthetic lethal interactions with MYC underscores the essentiality of this pathway for MYC function. Thus, nuclear export of RNAs is a dependency of MYC-driven cancers.”

REVIEWERS' COMMENTS

Reviewer #1 (Remarks to the Author):

The authors have done extensive work to thoroughly revise the manuscript as per our comments. Overall, the manuscript is significantly improved and we are satisfied with this version. We only recommend the following minor textual change to enhance the manuscript a little further:

In their response to the comment 1.3, the authors propose that the differences in efficacies observed between murine and PDX models could potentially be attributed to the modulation of the tumour microenvironment by XPO1 inhibition. It would be good to mention this in the discussion, especially since the authors do mention a possibility of future studies on MYC-SL and host immune response.

Reviewer #2 (Remarks to the Author):

The resubmitted manuscript answered most of our questions and the quality of the manuscript has been improved. The results in the paper support that, "Nuclear to Cytoplasmic Transport is a Druggable Dependency in MYC-Driven Hepatocellular Carcinoma", which is an interesting and important finding. The pathways analysis showed many nuclear export genes are induced by MYC and required in MYC-high states. There is extensive validation with XPO1. We have relatively minor comments regarding the screening results for XPO1.

XPO1 may have been selected for further validation because existing drugs can target this gene. The revision shows convincingly that drugs or siRNAs targeting XPO1 can increase cell death in MYC-high cells. Other genes might be predicted to have even greater synthetic lethality with MYC expression based on the screening results, but might not be good drug targets.

In the first part of our query 1 from the initial review, we wanted to see an analysis of the read counts (or normalized read counts) for each of the gRNAs targeting XPO1 at baseline and after selection in the MYC-High and MYC-low conditions. This is still missing in this version.

The revision is responsive to the second part of query 1 with a newly added Figure S6. However, the distribution of XPO1 in this chart does not look too promising since it is depleted in both MYC-high and MYC-low cells. We guess that in the MYC-low group the FDR did not reach <0.05 as it did in the MYC-

high group, which is the reason XPO1 falls in the MYC-SL category. But relative to the MYC-low cells, the MYC-high cells may have only marginally greater depletion.

Regardless of whether the screen showed only marginal synthetic lethality for XPO1, the human data from TCGA and the validation in vitro and in vivo were strong, and we agree with the conclusion that in MYC-high states there is a synthetic lethal requirement for nuclear export genes as a class. We support the publication of this manuscript.

Reviewer #3 (Remarks to the Author):

The manuscript by Deutzmann et al. is much improved in terms of presentation and clarity, which I expect will make the findings more accessible to readers from a broad audience. The expanded discussion of MYC-SL hits involving RNA transport genes also significantly broadens the interest of the paper. There are still a few issues with the text that I suggest be changed to fairly present the findings and current knowledge, which are:

1. Line 13 states, "We infer that MYC may generally regulate and require expression of nucleocytoplasmic transport genes for tumorigenesis". That sentence seems to indicate that nucleocytoplasmic transport genes may not be required in other instances, but given that many of these genes are essential for life, the sentence should be changed. I would suggest "We infer that MYC may generally regulate and require altered expression of nucleocytoplasmic transport genes for tumorigenesis".
2. Line 141 states, "XPO1 is an export receptor that forms a complex with Ran GTPase and drives the transport of multiple RNA and protein targets". I think this gives a false sense of the importance of XPO1/CRM1 in protein transport. I would suggest "XPO1 is an export receptor that forms a complex with Ran GTPase and drives the transport of multiple classes of RNA and >1000 known protein targets".
3. Line 167 states, "To further explore the role of RNA transport in MYC-driven cancer, we examined the effect of XPO1 inhibition by Selinexor...". The issue here is that this drug generally inhibits RNA and protein transport, which this sentence does not reflect. As such, I would suggest "To further explore the role of nucleocytoplasmic transport in MYC-driven cancer, we examined the effect of XPO1 inhibition by Selinexor...".

4. The sentence on line 180 has the same issue as point #3. The sentence should read “...therapeutic inhibition of the nucleocytoplasmic transport gene...”.

5. Line 222 states, “Thus, MYC is a key regulator of nuclear to cytoplasmic transport”. I do not think this statement can be made based on the data presented. It is a possibility, but from proven, so I suggest this sentence is deleted.

6. Line 254 states, “We show here that the inhibition of XPO1 with Selinexor blocked RNA transport...”. Again Selinexor will also block protein transport, so this aspect also needs to be conveyed or the reference to blocking RNA export removed.

7. Line 274 states, “Thus, nuclear export of RNAs is a dependency of MYC-driven cancers”. As with point #1, the sentence seems to indicate that somethings are not dependent on RNA export, which is an essential process. I would suggest ““Thus, altered nuclear export of RNAs is a potential dependency of MYC-driven cancers””.

RESPONSE TO REVIEWERS' COMMENTS

We thank the reviewers for their feedback and suggestions how to further improve our manuscript. Our direct response to the reviewers' comments is highlighted in blue and we use line numbers of the new draft and quotations to refer to changes we have made. In the revised manuscript, all edits are tracked.

REVIEWERS' COMMENTS

Reviewer #1 (Remarks to the Author):

The authors have done extensive work to thoroughly revise the manuscript as per our comments. Overall, the manuscript is significantly improved and we are satisfied with this version. We only recommend the following minor textual change to enhance the manuscript a little further:

In their response to the comment 1.3, the authors propose that the differences in efficacies observed between murine and PDX models could potentially be attributed to the modulation of the tumour microenvironment by XPO1 inhibition. It would be good to mention this in the discussion, especially since the authors do mention a possibility of future studies on MYC-SL and host immune response.

We thank the reviewer for this feedback and have now amended the discussion as follows:

Lines 304-308: “Among the advantages of our model is that it is fully immune competent. Drug efficacies could be underestimated in immunocompromised xenograft-based models which lack the contribution of an anti-cancer immune response. For instance, inhibiting a MYC-SL target may also have beneficial on-target but off-tumor effects on the tumor microenvironment. A recent example is the stimulation of anti-tumor immunity by BRD4 inhibition⁷⁸⁻⁸⁰.”

Reviewer #2 (Remarks to the Author):

The resubmitted manuscript answered most of our questions and the quality of the manuscript has been improved. The results in the paper support that, “Nuclear to Cytoplasmic Transport is a Druggable Dependency in MYC-Driven Hepatocellular Carcinoma”, which is an interesting and important finding. The pathways analysis showed many nuclear export genes are induced by MYC and required in MYC-

high states. There is extensive validation with XPO1. We have relatively minor comments regarding the screening results for XPO1.

XPO1 may have been selected for further validation because existing drugs can target this gene. The revision shows convincingly that drugs or siRNAs targeting XPO1 can increase cell death in MYC-high cells. Other genes might be predicted to have even greater synthetic lethality with MYC expression based on the screening results, but might not be good drug targets.

In the first part of our query 1 from the initial review, we wanted to see an analysis of the read counts (or normalized read counts) for each of the gRNAs targeting XPO1 at baseline and after selection in the MYC-High and MYC-low conditions. This is still missing in this version.

We thank the reviewer for this comment and apologize for this oversight of including the analysis of the read counts. We have now included the following figure illustrating frequencies of sgRNAs targeting Xpo1 and Prmt5 for baseline, MYC^{high}, and MYC^{low} conditions in **Supplementary Figure 6b**. As the reviewer points out below, the FDR of the Xpo1 Beta score in MYC^{low} did not reach <0.05 in the MYC^{low} condition. Xpo1 was thus considered essential in MYC^{high} (beta score: -0.22105, FDR: 0.038321) but not MYC^{low} cells (beta score: -0.13437, FDR: 0.47016).

The revision is responsive to the second part of query 1 with a newly added Figure S6. However, the distribution of XPO1 in this chart does not look too promising since it is depleted in both MYC-high and MYC-low cells. We guess that in the MYC-low group the FDR did not reach <0.05 as it did in the MYC-high group, which is the reason XPO1 falls in the MYC-SL category. But relative to the MYC-low cells, the MYC-high cells may have only marginally greater depletion.

Regardless of whether the screen showed only marginal synthetic lethality for XPO1, the human data from TCGA and the validation in vitro and in vivo were strong, and we agree with the conclusion that in MYC-high states there is a synthetic lethal requirement for nuclear export genes as a class. We support the publication of this manuscript.

We

Kirk J. Wangensteen, MD/PhD

Dingzi Yin, PhD

Mayo Clinic

Reviewer #3 (Remarks to the Author):

The manuscript by Deutzmann et al. is much improved in terms of presentation and clarity, which I expect will make the findings more accessible to readers from a broad audience. The expanded discussion of MYC-SL hits involving RNA transport genes also significantly broadens the interest of the paper. There are still a few issues with the text that I suggest be changed to fairly present the findings and current knowledge, which are:

We appreciate the reviewer's thoughtful feedback which helped us to refine our phrasing and improve accuracy. We have now included several changes to the text according to the reviewer's suggestion.

1. Line 13 states, "We infer that MYC may generally regulate and require expression of nucleocytoplasmic transport genes for tumorigenesis". That sentence seems to indicate that nucleocytoplasmic transport genes may not be required in other instances, but given that many of these genes are essential for life, the sentence should be changed. I would suggest "We infer that MYC may generally regulate and require altered expression of nucleocytoplasmic transport genes for tumorigenesis".

We have changed this sentence to now state **(Lines 13-14)**: "We infer that MYC may generally regulate and require altered expression of nucleocytoplasmic transport genes for tumorigenesis."

2. Line 141 states, "XPO1 is an export receptor that forms a complex with Ran GTPase and drives the transport of multiple RNA and protein targets". I think this gives a false sense of the importance of XPO1/CRM1 in protein transport. I would suggest "XPO1 is an export receptor that forms a complex with Ran GTPase and drives the transport of multiple classes of RNA and >1000 known protein targets". We have changed this sentence to now state **(Line 143-145)**: „XPO1 is an export receptor⁴⁰⁻⁴⁴ that forms a complex with Ran GTPase and drives the transport of multiple classes of RNA and more than a thousand protein targets⁴⁵⁻⁴⁷.“

3. Line 167 states, "To further explore the role of RNA transport in MYC-driven cancer, we examined the effect of XPO1 inhibition by Selinexor...". The issue here is that this drug generally inhibits RNA and protein transport, which this sentence does not reflect. As such, I would suggest "To further explore the role of nucleocytoplasmic transport in MYC-driven cancer, we examined the effect of XPO1 inhibition by Selinexor...".

We have changed this sentence to now state **(Lines 170-172)**: "To further explore the role of nucleocytoplasmic transport in MYC-driven cancer, we examined the effect of XPO1 inhibition by Selinexor in a primary transgenic mouse model of HCC (LAP-tTA/tet-O-MYC)¹³."

4. The sentence on line 180 has the same issue as point #3. The sentence should read "...therapeutic inhibition of the nucleocytoplasmic transport gene...".

We have changed this sentence to now state **(Lines 183-186)**: "Thus, therapeutic inhibition of the nucleocytoplasmic transport gene, XPO1, induces tumor regression without affecting normal adjacent liver in an autochthonous transgenic mouse model of MYC-induced HCC."

5. Line 222 states, "Thus, MYC is a key regulator of nuclear to cytoplasmic transport". I do not think this statement can be made based on the data presented. It is a possibility, but from proven, so I suggest this sentence is deleted.

We have now deleted this sentence as suggested by the reviewer.

6. Line 254 states, “We show here that the inhibition of XPO1 with Selinexor blocked RNA transport...”. Again Selinexor will also block protein transport, so this aspect also needs to be conveyed or the reference to blocking RNA export removed.

We have specifically demonstrated the inhibition of RNA transport by Selinexor treatment *in vitro* as shown in Figure 3d. However, we acknowledge that Selinexor may also affect transport of proteins that are essential to MYC-driven proliferation as we point out in the following sentence. We now state (**Lines 259-262**): “We show here that inhibition of XPO1-mediated nucleocytoplasmic transport blocked growth *in vitro*, induced tumor regression *in vivo* in a transgenic mouse, and inhibited human HCC PDX, more effectively in MYC^{high} cancers.”

7. Line 274 states, “Thus, nuclear export of RNAs is a dependency of MYC-driven cancers”. As with point #1, the sentence seems to indicate that somethings are not dependent on RNA export, which is an essential process. I would suggest ““Thus, altered nuclear export of RNAs is a potential dependency of MYC-driven cancers””.

We have changed this sentence to now state (**Lines 279-280**): „Thus, altered nuclear export of RNAs is a dependency of MYC-driven cancers.”